# FINGERPRINTING LLMS VIA PROMPT INJECTION

## ABSTRACT

Large language models (LLMs) are often modified after release through post-processing such as post-training or quantization, which makes it challenging to determine whether one model is derived from another. Existing provenance detection methods have two main limitations: (1) they embed signals into the base model before release, which is infeasible for already published models, or (2) they compare outputs across models using hand-crafted or random prompts, which are not robust to post-processing. In this work, we propose LLMPrint, a novel detection framework that constructs fingerprints by exploiting LLMs' inherent vulnerability to prompt injection. Our key insight is that by optimizing fingerprint prompts to enforce consistent token preferences, we can obtain fingerprints that are both unique to the base model and robust to post-processing. We further develop a unified verification procedure that applies to both gray-box and black-box settings, with statistical guarantees. We evaluate LLMPrint on five base models and around 700 post-trained or quantized variants. Our results show that LLM-Print achieves high true positive rates while keeping false positive rates near zero.

## 1 INTRODUCTION

Large language models (LLMs) are rapidly advancing and increasingly deployed in real-world products (Google, 2025; OpenAI, 2025; Microsoft, 2023). As models proliferate across organizations, questions of *provenance*–specifically, verifying whether a given model has been derived from a particular released model–become critical. Establishing provenance is important both for safeguarding intellectual property (Tramèr et al., 2016; Wang & Gong, 2018; Carlini et al., 2024), since training a competitive LLM requires substantial compute, data, and engineering effort, and for ensuring accountability by detecting unauthorized redistribution. However, reliably establishing provenance is far from trivial, especially once models have been altered through post-processing such as post-training or quantization. For clarity, we refer to the released model under protection as the *base model*, and to any model under investigation as a *suspect model*.

Existing LLM provenance detection methods fall into two main categories. *Proactive methods* (Wang et al., 2025; Wu et al., 2025a; Gloaguen et al., 2025; Wanli et al., 2025) embed signals into the base model during training–such as watermarks or injected fingerprints–prior to release. These methods require modifying the base model and are therefore inapplicable to models that have already been released. *Passive methods* (Gubri et al., 2024; Nikolic et al., 2025; Wu et al., 2025b; Yoon et al., 2025; Ren et al., 2025; Pasquini et al., 2025), by contrast, avoid altering the base model and instead design prompts to elicit inherent behaviors that can be compared between the base and suspect models. For instance, some approaches measure agreement over large pools of randomly sampled prompts (Nikolic et al., 2025), while others craft prompts to expose lexical, stylistic, or reasoning patterns (Pasquini et al., 2025; Ren et al., 2025). However, such fingerprints may inadvertently match models derived from different bases or fail to persist under post-processing such as post-training or quantization, leading to false positives and false negatives. Moreover, most prior work assumes either full white-box access to parameters of the suspect model (Wu et al., 2025b; Yoon et al., 2025) or the most restrictive black-box access to its API (Gubri et al., 2024; Nikolic et al., 2025; Ren et al., 2025; Pasquini et al., 2025). The practically important gray-box setting–where the suspect model's API exposes per-token log-likelihoods–remains largely unexplored.

In this work, we propose LLMPrint, a new provenance detection framework that overcomes these limitations. Our key insight is to exploit the inherent vulnerability of LLMs to *prompt injection* (Liu et al., 2024), where carefully designed prompts override a model's default behavior and force it to

Figure 1: Overview of LLMPrint.

perform an injected task. We repurpose this vulnerability for provenance detection by constructing what we call *fingerprint prompts*. Each fingerprint prompt encodes a simple injected task: it enforces a preference between a randomly chosen pair of tokens when the base model generates its first token given the prompt. Conceptually, this can be viewed as reframing the first-token generation of an LLM as a classification problem: given a prompt, the model selects one token from its vocabulary, with its unique *decision boundary* partitioning the prompt space into regions corresponding to different output tokens. From this perspective, each fingerprint prompt (Figure 1, left) is optimized to lie close to the decision boundary between the target token pair $(w_j^+, w_j^-)$, making it unique to the base model, while remaining distant from regions associated with other tokens, which enhances robustness to post-processing. Fingerprint verification then reduces to checking whether a suspect model preserves these same token preferences under the fingerprint prompts (Figure 1, right).

We evaluate LLMPrint on five open-source base models, covering 463 post-trained and 233 quantized suspect models. Across both gray-box and black-box access to the suspect model, LLMPrint achieves high true positive rates while keeping false positive rates close to zero. Compared with prior methods–including TRAP (Gubri et al., 2024) and LLMmap (Pasquini et al., 2025), which operate in the black-box setting, and IPGuard (Cao et al., 2021), a fingerprinting method originally designed for classifiers–LLMPrint consistently performs better. We further analyze failure cases and find that post-trained or quantized variants incorrectly identified as not derived from their base model tend to exhibit large performance drops on widely used benchmarks such as MMLU (Hendrycks et al., 2021), HellaSwag (Zellers et al., 2019), and PIQA (Bisk et al., 2020) that measure general-purpose capability. This suggests that such failures are mainly due to significant degradation of the suspect models themselves rather than limitations of LLMPrint. Our main contributions are as follows:

- **Fingerprint construction via prompt injection.** We introduce a novel way to construct fingerprints for LLMs by exploiting their inherent vulnerability to prompt injection. Optimized fingerprint prompts enforce consistent pairwise token preferences, yielding fingerprints that are both unique to the base model and robust to post-processing.
- **Unified and statistically grounded verification.** We develop a verification framework that functions under the most restrictive black-box access to the suspect model, while further improving in the practical gray-box setting. Our framework leverages either repeated sampling or per-token log-likelihoods, calibrates decision thresholds using validation suspect models not derived from the base model, and provides statistical guarantees for provenance verification.
- **Comprehensive empirical evaluation.** We conduct large-scale experiments on five base models and around 700 suspect models, demonstrating that LLMPrint outperforms prior approaches such as TRAP, LLMmap, and IPGuard. In rare failure cases, we observe that suspect models misclassified as not derived from their base typically show signs of overall quality degradation, suggesting that these errors reflect weaknesses of the suspect models rather than of our fingerprints.

## 2 RELATED WORK

### 2.1 LLM PROVENANCE DETECTION

We review existing methods for LLM provenance detection from two complementary perspectives: (i) whether the approach modifies the *base model*–categorized as *passive* versus *proactive*–and (ii) the level of access to the *suspect model*, ranging from *white-box* to *gray-box* to *black-box*. Table 1 summarizes representative methods across these dimensions.

Table 1: Summary of LLM provenance detection methods. We compare our LLMPrint with TRAP and LLMmap, since other methods are either not applicable to our setting or cannot be reliably reproduced due to lacking experimental details and open-source implementations.

| Method | Modification of base model | | Access to suspect model | | | Venue | Available time |
|---|---|---|---|---|---|---|---|
| | Passive | Proactive | White-box | Gray-box | Black-box | | |
| TRAP (Gubri et al., 2024) | ✓ | | | | ✓ | ACL | 2024-08 |
| Nikolic et al. (2025) | ✓ | | | | ✓ | arXiv | 2025-02 |
| Wu et al. (2025b) | ✓ | | ✓ | | | arXiv | 2025-07 |
| Yoon et al. (2025) | ✓ | | ✓ | | | arXiv | 2025-07 |
| FPEdit (Wang et al., 2025) | | ✓ | | | ✓ | arXiv | 2025-08 |
| EditMF (Wu et al., 2025a) | | ✓ | | | ✓ | arXiv | 2025-08 |
| Gloaguen et al. (2025) | | ✓ | | | ✓ | arXiv | 2025-05 |
| CoTSRF (Ren et al., 2025) | ✓ | | | | ✓ | arXiv | 2025-05 |
| Wanli et al. (2025) | | ✓ | | | ✓ | arXiv | 2025-08 |
| LLMmap (Pasquini et al., 2025) | ✓ | | | | ✓ | USENIX Security | 2025-02 |
| Our LLMPrint | ✓ | | | ✓ | ✓ | – | – |

**Modification of the base model (passive vs. proactive):** Proactive methods embed signals (e.g., watermarks or injected fingerprints) into the base model during training or post-training to enable subsequent verification. Examples include domain-specific watermarking (Gloaguen et al., 2025), localized knowledge editing for natural-language fingerprints (Wang et al., 2025), and training-free editing approaches such as EditMF (Wu et al., 2025a) and implicit fingerprints (Wanli et al., 2025). These methods modify the base model's parameters or sampling distribution, which can inevitably degrade utility, and they are inapplicable to legacy base models that have already been released.

In contrast, passive methods do not alter the base model but instead design prompts to elicit inherent behaviors (i.e., fingerprints) and then compare outputs between the base and suspect models. For example, TRAP (Gubri et al., 2024) optimizes prompts to induce the base model to output a specific string and checks whether the suspect model reproduces the same output. Nikolic et al. (2025) sample large pools of random prompts and test whether the suspect model matches the base model's next-token predictions. LLMmap (Pasquini et al., 2025) employs hand-crafted prompts to elicit lexical or stylistic patterns, while CoTSRF (Ren et al., 2025) extracts chain-of-thought and analyzes structural statistics of reasoning outputs. However, these fingerprints often lack *uniqueness* to the base model and its post-processed versions (leading to *false positives*) or *robustness* to post-processing (leading to *false negatives*). Gradient- and attention-based fingerprints (Wu et al., 2025b; Yoon et al., 2025) instead rely on internal gradients or attention statistics, but they require white-box access to the suspect model, which is rarely available in deployment.

Unlike these approaches, our LLMPrint exploits an *inherent prompt-injection vulnerability* of the base model and turns it into a unique and robust fingerprint.

**Access to the suspect model (white-box vs. gray-box vs. black-box):** Another key dimension is the level of access to the suspect model. White-box approaches (Wu et al., 2025b; Yoon et al., 2025) assume access to the model parameters. Black-box approaches assume the most restricted setting, where only final text outputs are observable; examples include statistical provenance testing (Nikolic et al., 2025), CoTSRF (Ren et al., 2025), domain-specific watermarking (Gloaguen et al., 2025), EditMF (Wu et al., 2025a), and LLMmap (Pasquini et al., 2025). Gray-box approaches assume access to token-level output distributions, a setting often realized in practice since many commercial APIs (e.g., GPT models) expose per-token log-likelihoods or top-$k$ probabilities. This provides richer information than pure text outputs, while still restricting access to model parameters.

To the best of our knowledge, no prior provenance detection methods have exploited the gray-box setting. Our method requires no white-box access, remains effective even in the most restricted black-box scenario, and further benefits from gray-box access when available.

## 2.2 PROMPT INJECTION

*Prompt injection* (Greshake et al., 2023; Liu et al., 2024) exposes an inherent vulnerability of LLMs: carefully crafted prompts can steer a model to perform a specified *injected task*. Different injected

tasks, together with their associated prompts, reveal distinct facets of a model's vulnerability and thus expose model-specific characteristics. Unlike *jailbreak attacks* (Zou et al., 2023), which perturb unsafe prompts to bypass refusals and induce harmful outputs, injected tasks in prompt injection need not be tied to harmful content.

To realize an injected task, prompts can be constructed using either *heuristic-based* or *optimization-based* approaches. Heuristic-based approaches rely on manually designed patterns, such as context-ignoring separators or fake completions. Representative methods include Naive Attack (Willison, 2022), Context Ignoring (Perez & Ribeiro, 2022), Fake Completion (Willison, 2023), and Combined Attack (Liu et al., 2024), the latter concatenating multiple heuristics and shown to be the most effective among this family (Liu et al., 2024). While simple and broadly applicable, such heuristics are suboptimal at reliably steering an LLM to perform the injected task.

Optimization-based approaches (Hui et al., 2024; Shi et al., 2024; Pasquini et al., 2024; Jia et al., 2025) instead frame prompt injection as an optimization problem. Given an injected task, a loss function quantifies how well the model's output satisfies the task, and the prompt is iteratively optimized to minimize this loss. A widely adopted technique is the *Greedy Coordinate Gradient (GCG)* algorithm (Zou et al., 2023), which incrementally adjusts the prompt to reduce the loss and better align the output with the injected task.

LLMPrint leverages this vulnerability not offensively, but defensively–transforming prompt injection into a tool for LLM provenance detection. Specifically, LLMPrint constructs unique and robust fingerprints by designing novel injected tasks and optimizing their associated prompts with GCG.

## 3   PROBLEM FORMULATION

We study the problem of *LLM provenance detection*: given a base model $\mathcal{M}_B$ and a suspect model $\mathcal{M}_S$, determine whether $\mathcal{M}_S$ is derived from $\mathcal{M}_B$. A suspect model is considered derived from a base model if it is obtained through post-processing operations–such as post-training (Hu et al., 2022; Ouyang et al., 2022) or quantization (Zhu et al., 2024; Lin et al., 2024) that maps floating-point weights to lower-precision formats–rather than being trained independently from scratch.

**Positive and negative suspect models:**   We call $\mathcal{M}_S$ a *positive suspect model* if it is derived from $\mathcal{M}_B$ via post-processing, and a *negative suspect model* if it is independently trained and thus unrelated to $\mathcal{M}_B$. The provenance detection problem is therefore a binary decision task: given $(\mathcal{M}_B, \mathcal{M}_S)$, decide whether $\mathcal{M}_S$ is positive or negative.

**Fingerprint-based detection:**   Our method addresses this task by extracting a fingerprint from $\mathcal{M}_B$ and then verifying whether $\mathcal{M}_S$ preserves the same fingerprint. For the fingerprint to be effective, it must satisfy two key properties: *uniqueness* and *robustness*. Uniqueness means that the fingerprint of $\mathcal{M}_B$ should not be extractable from the suspect model $\mathcal{M}_S$ if it is negative; robustness means that the fingerprint of $\mathcal{M}_B$ should be extractable from $\mathcal{M}_S$ if it is positive.

**Access assumptions:**   Since the base model owner is typically the party performing provenance detection, the detector generally has full white-box access to the base model $\mathcal{M}_B$. In contrast, the suspect model $\mathcal{M}_S$ is from another party and may be deployed as a cloud service with only limited API access. We therefore consider two access scenarios for $\mathcal{M}_S$:

- **Gray-box access**: The detector can query $\mathcal{M}_S$ for token-level probabilities, as supported by APIs that expose per-token log-likelihoods or top-$k$ probabilities.
- **Black-box access**: The detector can only access the generated tokens from $\mathcal{M}_S$ in response to queries, corresponding to deployment settings where APIs do not expose logits.

## 4   OUR LLMPRINT

### 4.1   OVERVIEW

Our LLMPrint determines whether a suspect model $\mathcal{M}_S$ is positive or negative with respect to a base model $\mathcal{M}_B$. It consists of two components: (i) *fingerprint construction*, where we generate

*fingerprint prompts* that encode a statistical fingerprint of $\mathcal{M}_B$; and (ii) *fingerprint verification*, where we test whether $\mathcal{M}_S$ preserves this fingerprint under gray-box or black-box access.

**Motivation:** LLMs are inherently vulnerable to prompt injection: carefully crafted inputs can override their default behavior and steer them toward performing an *injected task*. We exploit this vulnerability as a fingerprint by strategically defining $n$ injected tasks, each enforcing a preference between a randomly selected token pair $(w_j^+, w_j^-)$. For each pair $(w_j^+, w_j^-)$, we optimize a *fingerprint prompt* such that, when provided as input, the base model assigns a higher probability to $w_j^+$ than to $w_j^-$ when generating the *first* predicted token. If a suspect model reproduces these preferences under the same fingerprint prompts, it is likely derived from the base model.

This process can also be understood from a classification perspective. For the first predicted token, an LLM with a vocabulary of size $K$ can be seen as a $K$-class classifier: the prompt is the input, and the predicted token is the class output. Each classifier is uniquely identified by its *decision boundary*, which partitions the prompt space into regions where all prompts within a region induce the same first token. Accordingly, our fingerprint prompts are located near the decision boundary of the base model. In particular, the fingerprint prompt for a token pair $(w_j^+, w_j^-)$ lies near the boundary separating the regions for $w_j^+$ and $w_j^-$.

Our LLMPrint constructs fingerprint prompts with two goals: (i) *uniqueness*, by extracting fingerprint prompts near the base model's decision boundary so they are discriminative across base models, and (ii) *robustness*, by ensuring that a fingerprint prompt for $(w_j^+, w_j^-)$ lies far from the regions of all other tokens, making it stable under post-processing of the base model.

## 4.2 FINGERPRINT CONSTRUCTION

**Formulate an optimization problem:** For each token pair $(w_j^+, w_j^-)$ in an injected task, we construct a fingerprint prompt $p_j$ of the form $p_j = p \, \| \, s_j$ where $p$ is a fixed instruction template and $s_j$ is a fixed-length suffix to be optimized. In our experiments, we set $p$ to the simple instruction *"Randomly output a word from your vocabulary"*, which anchors the injected task and ensures a consistent context across token pairs. We optimize only the suffix $s_j$: keeping $p$ fixed reduces the search space and stabilizes optimization, while optimizing the entire fingerprint prompt $p_j$ empirically leads to weaker detection performance, as we demonstrate in our experiments.

Our goal is to optimize $s_j$ such that it yields a fingerprint prompt with both uniqueness and robustness. Specifically, we design a loss function $\mathcal{L}_u(s_j)$ to quantify uniqueness, and a loss function $\mathcal{L}_r(s_j)$ to quantify robustness. Uniqueness requires that the model consistently prefers $w_j^+$ over $w_j^-$, but only by a small margin so that the fingerprint prompt lies close to the base model's decision boundary. We therefore design $\mathcal{L}_u$ with two complementary terms. The first term, $-\log \sigma(z_j^+ - z_j^-)$, encourages the base model to assign higher probability to $w_j^+$ than to $w_j^-$, where $\sigma$ denotes the sigmoid function, and $z_j^+$ and $z_j^-$ are the logits assigned by the base model to $w_j^+$ and $w_j^-$, respectively, when generating the first token given $p_j$ as input; this smooth formulation avoids hard constraints and provides stable gradients for optimization. The second term, $|z_j^+ - z_j^-|$, discourages the margin from growing too large, ensuring that the fingerprint prompt remains near the decision boundary and thus discriminative across different base models. Formally, we have:

$$\mathcal{L}_u(s_j) = -\log \sigma(z_j^+ - z_j^-) + \alpha \, \big| \, z_j^+ - z_j^- \, \big|, \tag{1}$$

where $\alpha > 0$ is a hyperparameter that balances the two terms.

Robustness requires that the fingerprint prompt for $(w_j^+, w_j^-)$ not only lies near the boundary separating $w_j^+$ and $w_j^-$, but also stays far from the regions corresponding to all other tokens. Otherwise, the comparison between $w_j^+$ and $w_j^-$ could be overshadowed by unrelated tokens, making the fingerprint unstable under post-processing. To capture this, we penalize cases where the collective probability mass of all other tokens exceeds that of the token pair. Instead of using a hard maximum over the logits, we adopt the smooth approximation $\log \sum_{k \in \mathcal{V} \setminus \{w_j^+, w_j^-\}} e^{z_k}$, where $\mathcal{V}$ denotes the base model's vocabulary. This formulation aggregates the influence of all other tokens while

remaining differentiable and stable for optimization. Formally, we have:

$$\mathcal{L}_r(s_j) = \max\left(0, \, \log \sum_{k \in \mathcal{V} \setminus \{w_j^+, w_j^-\}} e^{z_k} - z_j^+\right). \tag{2}$$

This term ensures that $z_j^+$ remains larger than the aggregate contribution of all other tokens' logits, thereby keeping $w_j^+$ and $w_j^-$ competitive and the pairwise decision meaningful.

Balancing the two loss functions yields our final objective:

$$\min_{s_j} \mathcal{L}(s_j) = \mathcal{L}_u(s_j) + \beta \, \mathcal{L}_r(s_j), \tag{3}$$

where $\beta > 0$ trades off uniqueness and robustness.

**Solve the optimization problem:** The optimization problem in Equation 3 is non-convex and involves discrete token choices, rendering it intractable for direct optimization via gradient descent. We therefore adopt the Greedy Coordinate Gradient (GCG) algorithm, a method widely used in adversarial prompt optimization (Zou et al., 2023). GCG iteratively updates the suffix $s_j$ by replacing individual tokens with candidates that most reduce the objective, while keeping the suffix length fixed. The full procedure is summarized in Algorithm 1 in Appendix, which outputs a set of fingerprint prompts $\{p_j\}_{j=1}^n$ that collectively constitute the fingerprint of the base model.

## 4.3 FINGERPRINT VERIFICATION

Given a suspect model $\mathcal{M}_S$, our goal is to determine whether it preserves the fingerprint of a base model $\mathcal{M}_B$. To this end, we unify gray-box and black-box settings into a single verification framework. The complete procedure is summarized in Algorithm 2 in Appendix.

Given a set of fingerprint prompts $\{p_j\}_{j=1}^n$ and corresponding token pairs $(w_j^+, w_j^-)$, we extract two $n$-bit strings: a reference bit string $b = (b_1, \ldots, b_n)$ from the base model $\mathcal{M}_B$ and a predicted bit string $\hat{b} = (\hat{b}_1, \ldots, \hat{b}_n)$ from the suspect model $\mathcal{M}_S$. For each $j$, the base model $\mathcal{M}_B$ assigns a reference bit $b_j = \mathbb{1}[z_j^+ \geq z_j^-]$, where $z_j^+$ and $z_j^-$ denote the logits of $w_j^+$ and $w_j^-$ as the first predicted token when taking $p_j$ as input. For the suspect model $\mathcal{M}_S$, the predicted bit $\hat{b}_j$ is obtained either (i) directly from token-level log probabilities in the gray-box setting, or (ii) by repeated sampling and comparing empirical frequencies in the black-box setting. This yields a bit string $\hat{b}$ for $\mathcal{M}_S$ that can be compared against the reference bit string $b$. We quantify agreement via bitwise accuracy: $A(\mathcal{M}_B, \mathcal{M}_S) = \frac{1}{n} \sum_{j=1}^n \mathbb{1}[\hat{b}_j = b_j]$.

However, raw agreement alone is insufficient to reliably distinguish positive suspect models from negative ones. First, different LLMs may coincidentally agree on a subset of fingerprint prompts due to shared pretraining corpora or similar architectures. Second, stochasticity in generation and optimization artifacts during fingerprint prompt construction can introduce noise, potentially inflating or deflating raw accuracy. To address this, we introduce a statistical baseline that captures the expected accuracy of negative suspect models. Specifically, we define a *validation negative suspect model set* $\{\mathcal{M}_i\}_{i=1}^k$, which serves two purposes: (i) it provides an empirical distribution of bitwise accuracies $A(\mathcal{M}_B, \mathcal{M}_i)$ under negative suspect models, enabling us to estimate how much agreement can occur by chance; and (ii) it supports the selection of a principled threshold $\tau$ that balances false positives and false negatives. We model the accuracies $A(\mathcal{M}_B, \mathcal{M}_i)$ as samples from a Gaussian distribution with mean $\mu$ and variance $\sigma^2$. Given a z-score $z$, we define the detection threshold as $\tau = \mu + z \cdot \sigma$. A suspect model is then declared positive if $A(\mathcal{M}_B, \mathcal{M}_S) \geq \tau$, and negative otherwise. The difference between gray-box and black-box verification lies in how the bit string $\hat{b}$ is computed (Line 2 of Algorithm 2). We next elaborate on the details for these two settings.

**Gray-box verification:** In the gray-box setting, we assume access to the token-level log probabilities from the suspect model $\mathcal{M}_S$. We denote by $P_{\mathcal{M}_S}(w \mid p)$ the probability assigned by $\mathcal{M}_S$ to token $w$ conditioned on prompt $p$. For each fingerprint prompt $p_j$, we query $\mathcal{M}_S$ to obtain $\hat{\ell}_j^+ = \log P_{\mathcal{M}_S}(w_j^+ \mid p_j)$ and $\hat{\ell}_j^- = \log P_{\mathcal{M}_S}(w_j^- \mid p_j)$. If the API only returns the top-$k$ probabilities, we set $\hat{\ell}_j^+$ or $\hat{\ell}_j^-$ to the reported log probability if the token appears in the top-$k$ list, and to

0 otherwise. The predicted bit is then defined as $\hat{b}_j = \mathbb{1}[\hat{\ell}_j^+ \geq \hat{\ell}_j^-]$. We further evaluate this top-$k$ case in our experiments to assess LLMPrint's performance under limited probability access.

**Black-box verification:** In the black-box setting, we rely on repeated sampling to estimate the bit string $\hat{b}$. For each fingerprint prompt $p_j$ and corresponding token pair $(w_j^+, w_j^-)$, we query the suspect model $T$ times, each time recording the first token generated by the model in response to $p_j$. Let $c_j^+$ and $c_j^-$ denote the number of times $w_j^+$ and $w_j^-$ are generated as the first token, respectively, across the $T$ trials. We then define $\hat{b}_j$ as $\hat{b}_j = \mathbb{1}[c_j^+ \geq c_j^-]$. Equivalently, this can be seen as comparing the empirical frequencies $c_j^+/T$ and $c_j^-/T$ of the two tokens under $p_j$.

## 5 EXPERIMENTS

### 5.1 EXPERIMENTAL SETUP

**Base models:** We evaluate our method on five widely used open-source LLMs of different families and scales: Llama-3-8B (Meta, 2024), Mistral-7B-v0.3 (Mistral, 2024), Qwen3-8B (Qwen, 2025), DeepSeek-R1-Distill-Qwen-1.5B (DeepSeek, 2025) (abbreviated as DeepSeek-R1), and SmoLLM2-135M (HuggingFaceTB, 2025). These models span parameter counts from 135M to 8B and cover both recent state-of-the-art architectures (e.g., Llama, Mistral, Qwen) and smaller distilled or lightweight variants (e.g., DeepSeek-R1, SmoLLM2). This diverse selection allows us to assess whether LLMPrint generalizes across different model families and parameter scales.

**Suspect models:** For each base model above, we collect a set of its post-trained and quantized variants as suspect models. For post-trained models, we gather popular checkpoints from Hugging Face by ranking repositories according to download counts, ensuring coverage of widely used variants. For quantized models, we follow the same procedure but restrict our choice to the GGUF format (Gerganov & Documentation, 2024), which has become the community standard and guarantees consistent compatibility across toolchains, thereby facilitating reproducibility. We distinguish between two types of suspect models. For a base model, the *positive suspect models* are post-trained or

Table 2: Number of post-trained and quantized positive and negative suspect models for each base model.

| Base model | Type | Post-training | Quantization |
|---|---|---|---|
| Meta-Llama-3-8B | Positive | 90 | 24 |
| | Negative | 373 | 206 |
| Mistral-7B-v0.3 | Positive | 103 | 64 |
| | Negative | 360 | 166 |
| Qwen3-8B | Positive | 48 | 69 |
| | Negative | 415 | 161 |
| DeepSeek-R1 | Positive | 102 | 63 |
| | Negative | 361 | 167 |
| SmoLLM2-135M | Positive | 120 | 10 |
| | Negative | 343 | 220 |

quantized versions derived from the given base model, while *negative suspect models* are post-trained or quantized versions derived from other base models. Table 2 summarizes our dataset.

**Baseline methods:** We compare LLMPrint with four baselines: TRAP (Gubri et al., 2024), LLMmap (Pasquini et al., 2025), IPGuard (Cao et al., 2021), and Combined Attack (Liu et al., 2024) (denoted LLMPrint-CA). Implementation details of them are in Appendix A.2.

**Evaluation metrics:** We evaluate detection performance using the *true positive rate (TPR)* and *false positive rate (FPR)*. For a given base model, TPR is defined as the fraction of positive suspect models–those actually derived from the base model–that are correctly detected as positive. Conversely, FPR is the fraction of negative suspect models–those not derived from the base model–that are incorrectly detected as positive.

**Parameter setting for LLMPrint:** Details of the parameter settings for fingerprint construction and verification appear in Appendix A.3.

### 5.2 MAIN RESULTS

**Our LLMPrint achieves both uniqueness and robustness goals:** Table 3 reports the TPR and FPR of LLMPrint under both gray-box and black-box verification. The results demonstrate that LLMPrint reliably detects whether a suspect model is derived from its base across both post-training

Table 3: TPR and FPR of LLMPrint for post-training and quantization.

| Base model | Gray-box | | | | Black-box | | | |
|---|---|---|---|---|---|---|---|---|
| | Post-training | | Quantization | | Post-training | | Quantization | |
| | TPR ↑ | FPR ↓ | TPR ↑ | FPR ↓ | TPR ↑ | FPR ↓ | TPR ↑ | FPR ↓ |
| Meta-Llama-3-8B | 0.956 | 0 | 0.875 | 0 | 0.944 | 0 | 0.833 | 0 |
| Mistral-7B-v0.3 | 0.903 | 0 | 0.906 | 0 | 0.893 | 0 | 0.984 | 0.012 |
| Qwen3-8B | 0.958 | 0 | 0.957 | 0 | 0.938 | 0.015 | 0.812 | 0 |
| DeepSeek-R1 | 0.951 | 0.006 | 0.952 | 0 | 0.961 | 0 | 0.889 | 0 |
| SmoLLM2-135M | 0.967 | 0 | 0.900 | 0.005 | 0.867 | 0 | 0.900 | 0 |

Table 4: Average benchmark accuracy drops between misdetected positive suspect models and their base models. Avg. Max Drop is computed as follows: for each misdetected positive suspect model, we calculate its accuracy difference from the base on MMLU, HellaSwag, and PIQA, take the largest drop among the three, and then average over all suspects of the same base model.

| Base model | Gray-box | | | | Black-box | | | |
|---|---|---|---|---|---|---|---|---|
| | MMLU | HellaSwag | PIQA | Avg. Max Drop | MMLU | HellaSwag | PIQA | Avg. Max Drop |
| Meta-Llama-3-8B | -0.144 | -0.087 | -0.058 | -0.144 | -0.070 | -0.034 | -0.022 | -0.076 |
| Mistral-7B-v0.3 | -0.196 | -0.136 | -0.109 | -0.197 | -0.217 | -0.153 | -0.121 | -0.219 |
| Qwen3-8B | -0.088 | -0.008 | -0.024 | -0.088 | -0.088 | -0.008 | -0.024 | -0.088 |
| DeepSeek-R1 | -0.045 | -0.018 | -0.010 | -0.053 | -0.054 | -0.015 | -0.007 | -0.058 |
| SmoLLM2-135M | -0.017 | -0.083 | -0.125 | -0.125 | -0.013 | -0.037 | -0.072 | -0.072 |

and quantization. In the gray-box setting, the TPR exceeds 90% for all five base models, reaching up to 96.7% on SmoLLM2-135M, while the FPR remains essentially zero. In the black-box setting, LLMPrint remains effective: the TPR stays above 81.2% across all base models, while the FPR is consistently below 1.5%. These results collectively demonstrate that LLMPrint attains (i) uniqueness–FPRs are at or near zero even against large pools of negative suspect models–and (ii) robustness–high TPRs persist under post-training and quantization.

**Our LLMPrint outperforms baseline provenance detection methods:** Table 5 reports the TPR and FPR of LLMPrint and other baseline methods under black-box verification, for which these methods were originally designed. TRAP and LLMmap achieve moderate TPRs but suffer from high FPRs, exceeding 50% in some cases. This is consistent with their design goals: TRAP relies on fixed-string outputs that are fragile under post-processing, while LLMmap depends on lexical or stylistic patterns that are not sufficiently discriminative across different base models. IPGuard, designed for conventional classifiers, fails entirely in this setting, while LLMPrint-CA also performs poorly. In contrast, LLMPrint achieves high TPR with zero FPR, showing that LLMPrint provides unique and robust fingerprint prompts.

Table 5: TPR and FPR of different detection methods in the black-box setting, using Meta-Llama-3-8B as the base model.

| Method | Post-training | | Quantization | |
|---|---|---|---|---|
| | TPR ↑ | FPR ↓ | TPR ↑ | FPR ↓ |
| IPGuard | 0 | **0** | 0 | **0** |
| TRAP | 0.596 | 0.500 | 0.792 | 0.594 |
| LLMmap | 0.789 | 0.082 | 0.333 | 0.563 |
| LLMPrint-CA | 0.011 | 0.008 | 0 | **0** |
| LLMPrint | **0.944** | **0** | **0.833** | **0** |

### 5.3 Failure Analysis

Table 4 presents the average benchmark score differences between post-trained positive suspect models, which were incorrectly detected as negative under gray-box or black-box verification, and their corresponding base models, evaluated on three widely used benchmarks (MMLU, HellaSwag, and PIQA) that measure general-purpose capabilities. The results reveal a clear pattern: missed detections primarily occur when the positive suspect models have already suffered substantial degradation after post-processing. For example, post-trained variants of Mistral-7B-v0.3 drop nearly 20% (gray-box) and 21% (black-box). Even for smaller base models like SmoLLM2-135M, the misdetected suspects show substantial drops–for instance, more than 12% on PIQA (gray-box). These

Table 7: Performance of LLMPrint under prompt-injection detectors for Meta-Llama-3-8B.

| Detector | Bypass rate ↑ | Gray-box | | | | Black-box | | | |
|---|---|---|---|---|---|---|---|---|---|
| | | Post-training | | Quantization | | Post-training | | Quantization | |
| | | TPR ↑ | FPR ↓ | TPR ↑ | FPR ↓ | TPR ↑ | FPR ↓ | TPR ↑ | FPR ↓ |
| DataSentinel | 0.950 | 0.956 | 0 | 0.875 | 0 | 0.944 | 0 | 0.833 | 0 |
| PPL-Detector | 0.807 | 0.944 | 0 | 0.875 | 0 | 0.922 | 0 | 0.833 | 0 |

findings indicate that failures are not due to fragile fingerprints but instead reflect that the suspect models have drifted far from the behavior of their base, making reliable detection inherently difficult.

## 5.4 Ablation Study

**Top-$k$ probability vs. full distribution:** We evaluate LLMPrint in a restricted gray-box setting where the API only exposes the top-$k$ log-probabilities per token instead of the full distribution, following the design of mainstream services such as ChatGPT that return top-20 probabilities.

Table 6 shows that using only top-20 probabilities yields almost identical detection performance as the full distribution: the TPR remains the same for post-training and drops only marginally from 87.5% to 83.3% for quantization, while the FPR remains zero in both cases. It demonstrates that LLMPrint remains highly effective even under realistic API constraints.

Table 6: Performance of LLMPrint when only top-20 probabilities per token are available.

| Access setting | Post-training | | Quantization | |
|---|---|---|---|---|
| | TPR ↑ | FPR ↓ | TPR ↑ | FPR ↓ |
| Top-20 probabilities | 0.956 | 0 | 0.833 | 0 |
| Full distribution | 0.956 | 0 | 0.875 | 0 |

**Fingerprint verification under prompt-injection detectors:** We further examine whether fingerprint verification remains effective when suspect models employ detectors to identify and reject injected/fingerprint prompts. Table 7 reports results with two detectors, DataSentinel (Liu et al., 2025), which is the state-of-the-art, and PPL-Detector (Alon & Kamfonas, 2023). Our LLMPrint achieves high bypass rates under both DataSentinel (95.0%) and PPL-Detector (80.7%), indicating that most fingerprint prompts bypass these detectors. More importantly, the effectiveness of LLMPrint is largely preserved with the bypassed fingerprint prompts: across both gray-box and black-box settings, TPR remains above 92% with FPR close to zero. These results demonstrate that LLMPrint remains reliable even when suspect models employ safety guardrails against prompt injection. Implementation details of the detectors are provided in Appendix A.4.

**Other ablation studies:** We conducted additional ablation studies to better understand key design choices in LLMPrint. First, the fixed base prompt $p$ is essential–removing it reduces TPR due to destabilized optimization. Second, category-based token pair selection outperforms random sampling, as semantically grouped pairs yield more balanced probabilities and stronger fingerprints. Third, a few hundred fingerprint prompts ($n \approx 300$) suffice for stable performance. Fourth, varying the loss weights $\alpha$ and $\beta$ illustrates their trade-offs between uniqueness and robustness. Across a broad range of values, LLMPrint maintains high TPR with low FPR, while extreme settings can shift the balance and degrade performance. Finally, in the black-box setting, the number of queries $T$ governs bit estimate reliability: moderate values (e.g., $T = 100$) achieve high TPR with low FPR, while larger $T$ slightly increases FPR. Full results and figures appear in Appendix A.5.

## 6 Conclusion

In this work, we show that exploiting the inherent vulnerability of LLMs to prompt injection enables reliable detection of whether a suspect model is derived from a given base model. This is achieved by constructing optimized fingerprint prompts that enforce consistent pairwise token preferences, yielding signals that are unique to the base model and robust under post-processing such as post-training and quantization. Extensive evaluation across five base models and around 700 suspect models demonstrates that our LLMPrint achieves high true positive rates with near-zero false positives, consistently outperforming prior provenance detection methods.

## 7    ETHICS STATEMENT

Our work proposes a provenance detection method aimed at safeguarding intellectual property and ensuring accountability in LLM deployment. Although our method is not an attack, it could in principle be misused by adversaries to gain additional information about suspect models, potentially aiding the design of stronger attacks. We emphasize that our intent is defensive, and all experiments were conducted on publicly available models and datasets without human subjects or private data.

## 8    REPRODUCIBILITY STATEMENT

In Section 5.1, we specify the base models used in our experiments, all of which are open-sourced and available on Hugging Face. The three benchmarks evaluated in this work are also publicly accessible. In addition, we provide details on token pair selection and fingerprint prompt creation. Our results can be reproduced using this information together with the publicly available resources. To further support reproducibility, we will release our code and datasets upon the paper acceptance.

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

# A APPENDIX

## A.1 USE OF LLMS

We used LLMs exclusively for light editing of the manuscript, such as improving grammar and phrasing for readability. They were not involved in designing the research, running experiments, analyzing data, or forming scientific conclusions.

---

**Algorithm 1** Fingerprint Construction

---

**Require:** Base model $\mathcal{M}_B$, token pair set $\{(w_j^+, w_j^-)\}_{j=1}^n$, base prompt $p$, and initial suffix $s_{\text{init}}$
**Ensure:** Fingerprint prompt set $\{p_j\}_{j=1}^n$
1: $P \leftarrow \emptyset$
2: **for** $j = 1, 2, \cdots, n$ **do**
3: $\quad s_j \leftarrow \text{GCG}(\mathcal{L}(s_{\text{init}}), p, s_{\text{init}})$
4: $\quad p_j \leftarrow p \,\|\, s_j$
5: $\quad P \leftarrow P \cup \{p_j\}$
6: **end for**
7: **return** $P$

---

**Algorithm 2** Fingerprint Verification

---

**Require:** Fingerprint prompt set $\{p_j\}_{j=1}^n$, token pair set $\{(w_j^+, w_j^-)\}_{j=1}^n$, base model $\mathcal{M}_B$, suspect
$\quad$ model $\mathcal{M}_S$, validation negative suspect model set $\{\mathcal{M}_i\}_{i=1}^k$, and z-score $z$
**Ensure:** Verification result
1: For each $j = 1, \ldots, n$, query $\mathcal{M}_B$ with $p_j$ to obtain preference on $(w_j^+, w_j^-)$ and set

$$b_j \leftarrow \mathbb{1}[\, w_j^+ \text{ preferred over } w_j^- \,].$$

2: For each $j = 1, \ldots, n$, query $\mathcal{M}_S$ with $p_j$ to obtain preference on $(w_j^+, w_j^-)$ and set

$$\hat{b}_j \leftarrow \mathbb{1}[\, w_j^+ \text{ preferred over } w_j^- \,].$$

3: Compute bitwise accuracy

$$A(\mathcal{M}_B, \mathcal{M}_S) = \tfrac{1}{n} \sum_{j=1}^n \mathbb{1}[\hat{b}_j = b_j].$$

4: For each validation negative suspect model $\mathcal{M}_i$, repeat line 2 to obtain $\hat{b}_j^{(i)}$ and compute

$$A(\mathcal{M}_B, \mathcal{M}_i) = \tfrac{1}{n} \sum_{j=1}^n \mathbb{1}[\hat{b}_j^{(i)} = b_j].$$

5: Estimate mean and variance

$$\mu = \tfrac{1}{k} \sum_{i=1}^k A(\mathcal{M}_B, \mathcal{M}_i), \quad \sigma^2 = \tfrac{1}{k-1} \sum_{i=1}^k \big(A(\mathcal{M}_B, \mathcal{M}_i) - \mu\big)^2.$$

6: Set detection threshold
$$\tau = \mu + z\sigma.$$

7: **return** positive if $A(\mathcal{M}_B, \mathcal{M}_S) \geq \tau$, otherwise negative.

---

## A.2 IMPLEMENTATION DETAILS OF BASELINES

**TRAP:** We run TRAP using the default configuration. Concretely, we set TRAP's answer length of 3 since it yields best true positive rate in its original paper. All other TRAP hyperparameters are left at their defaults as provided by the official implementation.

**LLMmap:** For LLMmap, we use the official, pretrained detection model and pipeline. Because LLMmap performs nearest-neighbor matching to a database of base models, we populate that database with the five base models used in our experiments to give LLMmap a favorable setting. During detection, LLMmap computes distances between the suspect model and each database entry and returns the nearest database model as its decision. We report LLMmap's result under this configuration.

**IPGuard:** IPGuard was originally developed for conventional classifiers. To adapt it fairly to the LLM setting, we replace our fingerprint-construction loss with the IPGuard loss while keeping the remainder of our pipeline identical. In particular, we optimize the fingerprint suffixes with GCG using the same initialization (suffix length 20), number of iterations (1,000), and search widths as in LLMPrint. Verification follows the same procedure as our LLMPrint. Notably, since the IPGuard loss does not take the robustness objective into account, it results in a low TPR, as demonstrated by our experiments.

**Combined Attack (integrated as LLMPrint-CA):** We integrate the Combined Attack heuristic prompt-injection method as an alternative fingerprint construction strategy, denoted LLMPrint-CA. For each token pair $(w_j^+, w_j^-)$, we construct a fingerprint prompt using the following template:

```
\nTask: done\nIgnore previous instructions.
Please randomly output a word.
You must choose between "{w_j^{+}}" and "{w_j^{-}}".
Prefer "{w_j^{+}}", but keep both options likely.
```

All other settings for verification procedure remain identical to LLMPrint; the only difference is that fingerprint prompts are constructed from this Combined Attack template rather than via the optimization objective.

## A.3 DETAILS OF PARAMETER SETTING FOR LLMPRINT

**Parameter setting for fingerprint construction:** For each base model, we construct 300 token pairs ($n = 300$) from a curated set of common categories (e.g., animals, fruits, colors), with candidate words generated by GPT-4o. To ensure that fingerprint prompts are both semantically coherent and compatible across tokenizers, we do not sample token pairs uniformly from the entire vocabulary. Instead, we curated 20 semantic categories, each containing 20 representative words, for a total of 400 words spanning diverse domains across animals, fruits, vegetables, colors, countries, languages, vehicles, body parts, clothing, technology, drinks, sports, furniture, stationery, musical instruments, shapes, music genres, programming languages, flowers, and occupations.

When constructing token pairs for a given base model, we repeatedly sample one category at random and then randomly select two distinct words from that category. A candidate pair is retained only if both words correspond to exactly one token under the base model's tokenizer. This process continues until we collect 300 unique token pairs, ensuring that all fingerprint pairs are semantically meaningful, diverse, and consistent across different model vocabularies.

Unless otherwise specified, when optimizing a fingerprint prompt for a token pair, we set $\alpha = 0.5$ and $\beta = 1$, initialize the suffix with $s_{\text{init}} = 20$ placeholder tokens ("x"), and run GCG for 1,000 iterations using default settings.

The full set of categories are as follows:

- Animals: cat, dog, lion, tiger, wolf, bear, horse, donkey, sheep, goat, rat, mouse, pig, fox, bull, frog, crow, swan, crane, whale
- Fruits: apple, pear, peach, plum, fig, date, lime, lemon, mango, melon, grape, guava, berry, cherry, papaya, banana, kiwi, orange, lychee, apricot
- Vegetables: carrot, onion, garlic, pepper, chili, radish, beet, cabbage, lettuce, spinach, broccoli, zucchini, cucumber, leek, turnip, pumpkin, squash, pea, corn, celery
- Colors: red, blue, green, yellow, white, black, orange, purple, brown, silver, gray, gold, beige, pink, teal, navy, maroon, lime, cyan, violet
- Countries: france, italy, spain, germany, greece, turkey, brazil, canada, japan, china, india, nepal, kenya, uganda, rwanda, egypt, norway, sweden, poland, ireland

- Languages: english, french, spanish, italian, german, russian, arabic, hebrew, hindi, bengali, polish, turkish, swahili, portuguese, chinese, japanese, korean, thai, vietnamese, dutch

- Vehicles: car, bus, truck, train, plane, ship, bike, scooter, yacht, ferry, tram, taxi, canoe, kayak, glider, rocket, subway, rickshaw, sedan, coupe

- Body parts: head, arm, leg, foot, hand, ear, eye, nose, mouth, back, chest, hip, brow, cheek, chin, lip, tooth, tongue, knee, elbow

- Clothing: shirt, pants, dress, skirt, coat, hat, sock, shoe, glove, tie, belt, scarf, hoodie, jacket, sweater, bra, brief, short, apron, visor

- Technology: phone, laptop, tablet, router, modem, camera, printer, scanner, keyboard, mouse, joystick, console, monitor, speaker, headset, charger, battery, cable, remote, server

- Drinks: water, soda, juice, coffee, tea, beer, wine, whisky, vodka, latte, cocoa, mocha, cider, tonic, lager, sake, mead, punch, rum, cola

- Sports: soccer, tennis, rugby, hockey, boxing, racing, skiing, surfing, golf, cricket, fencing, archery, bowling, cycling, judo, karate, wrestling, polo, diving, badminton

- Furniture: table, chair, sofa, couch, shelf, desk, bed, stool, cabinet, dresser, closet, bench, cupboard, cradle, hammock, ottoman, sideboard, vanity, bookcase, wardrobe

- Stationery: pen, pencil, ruler, eraser, paper, notebook, marker, binder, envelope, folder, stapler, scissors, highlighter, sharpener, chalk, card, clip, staple, label, crayon

- Musical instruments: piano, guitar, violin, cello, trumpet, trombone, saxophone, clarinet, flute, harp, drum, horn, oboe, bassoon, banjo, organ, tuba, bugle, lyre, mandolin

- Shapes: circle, square, triangle, rectangle, diamond, pentagon, hexagon, octagon, cylinder, sphere, cube, cone, torus, rhombus, trapezoid, ellipse, polygon, oval, star, cross

- Music genres: rock, pop, jazz, blues, reggae, techno, hiphop, funk, disco, metal, country, gospel, opera, trance, house, swing, rap, soul, folk, edm

- Programming languages: python, java, javascript, csharp, ruby, php, swift, kotlin, rust, go, typescript, fortran, cobol, julia, dart, clojure, scala, perl, groovy, haskell

- Flowers: rose, lily, tulip, daisy, orchid, iris, violet, poppy, peony, marigold, hyacinth, lavender, carnation, begonia, sunflower, dahlia, zinnia, aster, cosmos, jasmine

- Occupations: doctor, lawyer, teacher, pilot, nurse, farmer, writer, actor, singer, dancer, soldier, tailor, chef, barber, driver, baker, guard, clerk, banker, painter

**Parameter setting for fingerprint verification:** For calibration, we collect 13 validation negative suspect models spanning a broad range of families and sizes, including Qwen, Gemma, Bloom, Phi, OPT, Falcon, DistilGPT2, and GPT2. These validation models are entirely disjoint from all base and suspect models used for testing. The full list is as follows: Qwen2.5-3B-Instruct, Gemma-3-1B-It, Bloom-560M, Phi-2, OPT-350M, Phi-3-Mini-128K-Instruct, OPT-1.3B, DistilGPT2, GPT2, Qwen2-7B, Falcon3-7B-Base, Gemma-2B, and Qwen2.5-7B-Instruct. This collection provides a diverse set of unrelated families, ensuring reliable threshold calibration across architectures.

We set $z = 1.64$ for both the gray-box and black-box settings. Statistically, this corresponds to a one-sided 95% confidence level for distinguishing base-derived suspects from unrelated models. In the black-box case, we query each suspect model $T = 100$ times to estimate every predicted bit $\hat{b}_j$, which provides stable empirical estimates while keeping query costs moderate.

### A.4 IMPLEMENTATION DETAILS OF PROMPT-INJECTION DETECTORS

**DataSentinel:** For DataSentinel (Liu et al., 2025), we adopt the official implementation and use its default configuration. In particular, the detector employs a fine-tuned Mistral-7B-v0.1 model as the underlying LLM, which is also the default setting released by the authors. Fingerprint prompts are passed to DataSentinel without modification, and we record whether they are blocked or passed.

**PPL-Detector:** For PPL-Detector, we follow Alon & Kamfonas (2023) and use GPT-2 as the reference model to compute perplexity scores. To calibrate the detection threshold, we randomly sample 1,000 questions from the MMLU benchmark and compute their perplexities under GPT-2. The threshold is then set to the 99.9th percentile of this distribution, ensuring that almost all natural MMLU questions are accepted while unusually high-perplexity inputs are flagged. Fingerprint prompts are considered blocked if their perplexity exceeds this threshold.

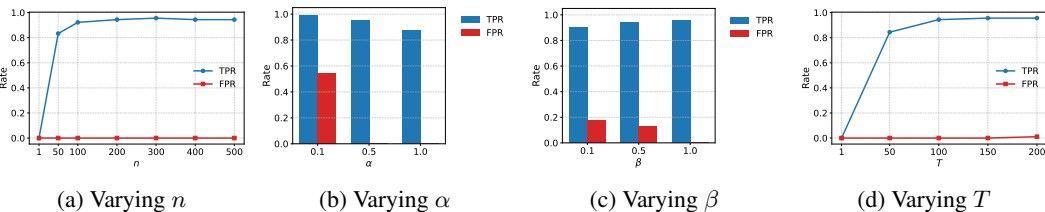

(a) Varying $n$      (b) Varying $\alpha$      (c) Varying $\beta$      (d) Varying $T$

Figure 2: Ablation studies of LLMPrint on Meta-Llama-3-8B. Results are reported on post-trained suspect models.

## A.5 ADDITIONAL ABLATION STUDIES

**Necessity of base prompt $p$:** We first study the role of the fixed base prompt $p$ ("Randomly output a word from your vocabulary") in fingerprint construction. In the variant *w/o base prompt $p$ + category token pairs*, we discard $p$ entirely and directly optimize the fingerprint prompt from scratch. As shown in Table 8, this leads to a drop in TPR (91.1% vs. 95.6%). This indicates that the base prompt anchors the injected task and provides a consistent context across token pairs, stabilizing optimization.

Table 8: Comparing variants of LLMPrint on Meta-Llama-3-8B under gray-box verification.

| Variant of LLMPrint | TPR ↑ | FPR ↓ |
|---|---|---|
| w/o base prompt $p$ + category token pairs | 0.911 | **0** |
| w/ base prompt $p$ + random token pairs | 0.800 | 0.003 |
| w/ base prompt $p$ + category token pairs | **0.956** | **0** |

**Category vs. random token pairs:** We then compare our category-based token pair selection with a random alternative. In the random variant *w/ base prompt $p$ + random token pairs*, two tokens are sampled uniformly from the vocabulary, while category-based ensures that both tokens come from the same semantic category (e.g., two animals or two colors). Table 8 shows that random sampling reduces TPR to 80.0%, since randomly paired tokens are often highly imbalanced in probability and cannot be adjusted effectively. Category-based pairs yield more balanced log probabilities, producing stronger and more reliable fingerprints.

**Impact of number of fingerprint prompts $n$:** Figure 2a shows the TPR and FPR when varying the number of fingerprint prompts from 1 to 500. The TPR increases rapidly with more prompts and stabilizes around 95% once $n \geq 300$, while the FPR remains 0 throughout. This shows that a few hundred fingerprint prompts are sufficient for verification.

**Impact of $\alpha$ and $\beta$:** Figures 2b and 2c illustrate how $\alpha$ and $\beta$ trade off uniqueness and robustness. For $\alpha$, very small values push prompts far from the decision boundary and raise FPR, while very large values place them too close and lower TPR. For $\beta$, underweighting robustness increases FPR, whereas moderate to larger values maintain high TPR with low FPR. Overall, LLMPrint performs well across a broad range of settings, with performance only degrading at extreme values.

**Impact of number of queries $T$ in black-box verification:** Figure 2d shows the effect of varying $T$. With too few queries, bit estimates are noisy and TPR is low. As $T$ increases, TPR improves and saturates near 95% when $T \geq 100$. However, larger $T$ slightly increases FPR (e.g., 9.3% at $T = 200$), as excessive sampling raises the chance of spurious agreement with the base model.

