# OpenReview forum: "Fingerprinting LLMs via Prompt Injection"
_ICLR.cc/2026/Conference — ICLR 2026 Conference Withdrawn Submission_

### Official Review · Reviewer_iVDu · 2025-10-14

**Soundness:** 2
**Presentation:** 1
**Contribution:** 3
**Rating:** 4
**Confidence:** 4

**Summary:**

LLMPrint aims to detect provenance of a given model passively, i.e., it does not require injecting or modifying the base model. For many token pairs ($w_j^+$, $w_j^-$), the authors optimize an injected prompt under the template “Randomly output a word from your vocabulary”, so that the base model mildly prefers $w_j^+$ for the first generated token, positioning each prompt near a decision boundary yet far from other tokens. The optimization process uses greedy coordinate gradient from LLM-attacks by Zou et al. The Verification compares the base’s bit string with a suspect model’s bit string (from gray-box log-probs or black-box sampling) using a statistically calibrated threshold. The paper reports experiments on five base models and different post-trained or quantized methods.

The method has made some strong assumptions, such as the base model and suspect models sharing the same token preferences under the fingerprint prompts. Instead of sampling from random prompts, the LLMPrint defines injected tasks where the model is enforced to present a preference between a selected token pair.

**Strengths:**

- Provenance verification is increasingly important as open-weight models proliferate and are post-processed (SFT/LoRA/quantization). This work advances passive testing without modifying the base model, a practical setup. Also, the assumption of black-box access to the suspect model is very practical.

- A comprehensive collection of very recent works in provenance detection, demonstrating a community interest, also motivating the work very well.

**Weaknesses:**

- The method’s decision-boundary intuition and the loss function (uniqueness + robustness) are not very clearly motivated. Is it universally agreed that the base model and suspect models share the same token preferences under the fingerprint prompts? Even two different base models can share the same preferences, right? Also, I am not convinced that "if a suspect model reproduces these preferences under the same fingerprint prompts, it is likely derived from the base model. I understand, "we introduce a statistical baseline that captures the expected accuracy of negative suspect models" which is related to solving this concern, but then the "an empiricial distribution" can be stochastic, as it varies along with the evaluated model sets.

- The core searching/optimization process uses GCG, which is already a widely-used method from 2023; I'm not sure about the technical novelty in this paper. The framework is from LLMmap and CoTSRF, but using GCG to find more effective prompts. The improved performance is built upon extra computing.

- The paper curates validation negatives and many test negatives but does not show open-world behavior where the suspect is none of the known families (or a strong instruction-tuned model that shares tokenizer/architecture). The strong near-zero FPRs could partly reflect distributional overlap between base/negatives seen during prompt optimization.

- "while negative suspect models are post-trained or quantized versions derived from other base models" my understanding is that all negative suspect models are from even different architectures and sizes? If so, the logit dimensions are already different.

**Questions:**

- There are five seed base models in total; all evaluation pairs are built upon the five?

- What is the intuition of "located near the decision boundary of the base model", when the model has a very "flat" predictive distribution (high entropy) over the vocabulary?

- except for "DeepSeek-R1-Distill-Qwen-1.5B", it is unclear whether all other base models are instruction versions or base versions. Will this impact the result? e.g., instruction models as base models are more difficult to detect?

- A distilled version of the model is not taken as a suspect model, right? Like, distilled DeepSeek is not taken as a suspect model of llama?

---

> ### Author Response · Authors · 2025-11-21
>
> We sincerely thank the reviewer for these encouraging comments, especially for recognizing the usefulness of supporting black-box suspect models, and the value of our comprehensive synthesis of recent provenance-detection research that highlights the community’s growing interest in this problem. Please find our detailed responses to your comments below.
>
> >**Weakness 1**: The method’s decision-boundary intuition and the loss function (uniqueness + robustness) are not very clearly motivated. Is it universally agreed that the base model and suspect models share the same token preferences under the fingerprint prompts? Even two different base models can share the same preferences, right? Also, I am not convinced that "if a suspect model reproduces these preferences under the same fingerprint prompts, it is likely derived from the base model. I understand, "we introduce a statistical baseline that captures the expected accuracy of negative suspect models" which is related to solving this concern, but then the "an empirical distribution" can be stochastic, as it varies along with the evaluated model sets.
>
> **Response**: We appreciate the reviewer’s questions. Our decision-boundary intuition follows from treating the LLM’s first-token prediction as a multi-class classifier over the vocabulary: under a given prompt, the model selects the token with high logit among a large number of competing classes. By optimizing fingerprint prompts such that two unrelated tokens become the top two classes with close logits, we effectively locate the model near a pairwise decision boundary. Crucially, our token pairs are sampled within broad semantic categories (colors, animals, fruits, etc.) but are not strongly correlated pairs like “good/great”, so there is no universal or linguistically determined ordering between them. Thus, any consistent preference between $w_j^+$​ and $w_j^-$​ arises from the model’s learned boundary geometry, not from inherent token bias.
>
> Regarding the concern about the empirical negative baseline being stochastic, we note that in our experiments all base models are evaluated against the same validation negative suspect model set, and we observe that the resulting distribution is highly stable, with variance far smaller than the separation between positive and negative bitwise accuracy. In practical deployments, the stability of the baseline can be further improved simply by enlarging the validation negative suspect model set, since the variance of the empirical null distribution decreases with sample size. Therefore, the baseline is not an arbitrary or fragile quantity. It converges toward a stable population-level estimate as more negative suspect models are included.
>
> >**Weakness 2**: The core searching/optimization process uses GCG, which is already a widely-used method from 2023; I'm not sure about the technical novelty in this paper. The framework is from LLMmap and CoTSRF, but using GCG to find more effective prompts. The improved performance is built upon extra computing.
>
> **Response**: We appreciate the reviewer’s comment. While LLMPrint uses GCG as its optimizer, GCG is only a generic optimization tool; the novelty of our method lies in the objective and problem formulation, not the optimizer. Unlike LLMmap or TRAP, which search for prompts that elicit specific outputs, LLMPrint introduces a decision-boundary–based fingerprinting formulation that explicitly optimizes for a pairwise token preference balancing uniqueness and robustness. Our experiments show that replacing our objective with a TRAP-style single-token objective, even under the same compute budget, leads to extremely high FPR (e.g., single-token FPR = 0.8633), demonstrating that the improvements do not come from “more computation” but from the boundary-based design itself. GCG simply serves as a convenient optimization backend; the technical contribution is the boundary-centered objective tailored for LLM fingerprinting, which to our knowledge has not appeared in prior work.

---

> ### Author Response · Authors · 2025-11-21
>
> >**Weakness 3**: The paper curates validation negatives and many test negatives but does not show open-world behavior where the suspect is none of the known families (or a strong instruction-tuned model that shares tokenizer/architecture). The strong near-zero FPRs could partly reflect distributional overlap between base/negatives seen during prompt optimization.
>
> **Response**: We appreciate the reviewer’s concern. Our setting already corresponds to the open-world scenario the reviewer described. We evaluate five different base models, and for each base model we optimize fingerprint prompts independently, without access to any information about the other base models or their suspect variants. In other words, the prompt optimization for a given base model is performed with no knowledge of the negative families used for the other bases, which is exactly the open-world condition where the detector has no prior knowledge about which negative suspect models may appear at test time. The near-zero FPRs therefore cannot be attributed to distributional overlap: the negatives used for testing a base model were never involved in its fingerprint optimization. We will clarify this point in the revision.
>
> >**Weakness 4**: "while negative suspect models are post-trained or quantized versions derived from other base models" my understanding is that all negative suspect models are from even different architectures and sizes? If so, the logit dimensions are already different.
>
> **Response**: We appreciate the reviewer’s question. In practical API-based verification (our threat model), the detector does not have access to the full logit vector or its dimensionality—many commercial APIs expose only the sampled token (black-box) or top-k log probabilities (gray-box). Therefore, differences in the underlying logit dimension or vocabulary size are not observable to the detector and cannot be used as a trivial discriminative signal. This is exactly why a behavioral fingerprint is needed: all suspect models, regardless of architecture, vocabulary size, or logit dimensionality, return outputs in the same unified token space (text strings), and our method operates solely on these observable outputs. Consequently, even if two models differ internally in logit dimension, this information is invisible in the API setting, and LLMPrint provides a non-trivial verification mechanism under this realistic constraint. We will clarify this point in the revision.
>
> >**Question 1**: There are five seed base models in total; all evaluation pairs are built upon the five?
>
> **Response**: Yes. We use five base models in total, and for each base model we independently construct its own 300 fingerprint token pairs. The fingerprint prompts for a given base model are optimized only with respect to that model and do not use any information from the other base models. This setup matches the open-world scenario where the detector does not know which negative models may appear at test time.
>
> >**Question 2**: What is the intuition of "located near the decision boundary of the base model", when the model has a very "flat" predictive distribution (high entropy) over the vocabulary?
>
> **Response**: The decision-boundary intuition does not rely on the model’s natural distribution being sharp. Even if the base model initially produces a flat, high-entropy distribution, the optimization explicitly reshapes the prompt so that the target tokens $w_j^+$ and $w_j^-$​ become the top two logits with very similar probabilities, while all other tokens remain much lower. In other words, GCG optimization forces a prompt that places the model in a boundary-adjacent region for that specific token pair, rather than assuming that such a boundary already exists in the model’s original predictive distribution.
>
> >**Question 3**: except for "DeepSeek-R1-Distill-Qwen-1.5B", it is unclear whether all other base models are instruction versions or base versions. Will this impact the result? e.g., instruction models as base models are more difficult to detect?
>
> **Response**: Our base model set already includes both base versions and instruction-tuned versions. Specifically, Qwen3-8B (used as one of our five seed base models) is an instruction model, while the other four base models are base (non-instruction) versions. Since LLMPrint achieves consistently high TPR on all five base models—including the instruction-tuned one—our results indicate that using an instruction model as the base does not negatively affect fingerprint detectability. Therefore, the concern that instruction models may be more difficult to detect is already addressed by our experimental setup.

---

> ### Author Response · Authors · 2025-11-21
>
> >**Question 4**: A distilled version of the model is not taken as a suspect model, right? Like, distilled DeepSeek is not taken as a suspect model of llama?
>
> **Response**: We would like to clarify the two possible interpretations of the question.
> (1) If “DeepSeek-R1-Distill-Llama-8B” is understood as a Llama-derived suspect model (i.e., a model obtained by post-training Llama using SFT or RL-style training), then this case is already covered in our evaluation. Our suspect model set contains many real-world SFT/RL-tuned variants of Llama-3-8B, and LLMPrint successfully detects the vast majority of them.
> (2) If the question instead refers to detecting whether a distilled version of DeepSeek-R1 is a suspect model of DeepSeek-R1 itself, then evaluating this requires running fingerprint construction of DeepSeek-R1, which is computationally beyond our available resources. The method is applicable in principle, and we view this as an important direction for future work.

---

### Official Review · Reviewer_n5a1 · 2025-10-27

**Soundness:** 2
**Presentation:** 2
**Contribution:** 2
**Rating:** 2
**Confidence:** 3

**Summary:**

This paper proposes a fingerprinting method via prompt injection to audit whether a suspect model is positive or negative with respect to a base model. Specifically, the authors optimize a suffix in the prompt to trigger a specific preference in the first-token generation process. They conduct extensive evaluations on over 700 post-trained or quantized model variants.

**Strengths:**

- Extensive evaluation, especially using so many model variants collected from HuggingFace.
- Strong performance.
- Well-structured paper.

**Weaknesses:**

- The paper claims novelty in using prompt injection for LLM fingerprinting, but this framing feels overstated. From the perspective of techniques, LLMPrint operates almost identically to prior adversarial optimization methods such as TRAP (and other adversarial attacks based method [a,b]), which also learn suffixes to control model behavior. The difference lies in controlling token-level preferences instead of full outputs—seems more like a minor objective change than a conceptual innovation. Moreover, prior adversarial methods could also be viewed as a form of prompt injection, since they induce non-native behaviors and override natural outputs. The established literature has described these as adversarial attacks, which is technically more accurate and avoids confusion with the security-oriented notion of prompt injection. This reframing seems to be a strategic rebranding of a well-understood technique for the sake of novelty. I would also encourage the authors to provide a more in-depth and technically grounded discussion in the related work section to clearly articulate how their approach truly differs from prior adversarial attack based methods.

- The authors claim their method's uniqueness comes from placing prompts near the decision boundary. However, this concept of near is ill-defined and controlled by a manually tuned hyperparameter \alpha (e.g., 0.5). How near is enough to be unique, yet far enough to be robust? The paper provides no theoretical analysis or empirical study on the sensitivity of \alpha. Without this, the central claim becomes arbitrary. It's possible the method works not because of the property of the decision boundary, but simply because the optimization finds a local minimum that happens to be transferable to its model variants, and the decision boundary is just a post-hoc rationalization.

- Although the experimental results show that LLMPrint substantially outperforms TRAP, the paper does not clearly explain why this is the case. Is the improvement solely due to controlling behavior at the first-token level, or are there deeper factors that make the proposed optimization more effective? It would be good if the authors provide stronger theoretical or analytical insights to justify why their method is inherently better than other adversarial-attack-based approaches.

-  The failure analysis in Section 5.3 attributes detection failures to a drop in the suspect model's benchmark performance. While the data shows a correlation, this attribution of causality is not sufficiently proven. The authors do not rule out confounding variables or the possibility that their fingerprints are simply not robust to the specific fine-tuning methods that also happen to degrade benchmark performance.

[a] UTF: Undertrained Tokens as Fingerprints A Novel Approach to LLM Identification

[b] ProFLingo: A Fingerprinting-based Intellectual Property Protection Scheme for Large Language Models

**Questions:**

- What are the differences between the proposed method and adversarial attack based methods?
- Why is the proposed method inherently better than other methods?
- Are the fingerprint prompts really near the decision boundary?

Please also see the above weaknesses.

---

> ### Author Response · Authors · 2025-11-21
>
> We sincerely thank the reviewer for highlighting these strengths, especially our extensive evaluation across a large number of real-world HuggingFace model variants, the strong empirical performance of our method, and the clear and well-structured presentation of the paper. Please find our detailed responses to your comments below.
>
> >**Weakness 1**: The paper claims novelty in using prompt injection for LLM fingerprinting, but this framing feels overstated. From the perspective of techniques, LLMPrint operates almost identically to prior adversarial optimization methods such as TRAP (and other adversarial attacks based method [a,b]), which also learn suffixes to control model behavior. The difference lies in controlling token-level preferences instead of full outputs—seems more like a minor objective change than a conceptual innovation. Moreover, prior adversarial methods could also be viewed as a form of prompt injection, since they induce non-native behaviors and override natural outputs. The established literature has described these as adversarial attacks, which is technically more accurate and avoids confusion with the security-oriented notion of prompt injection. This reframing seems to be a strategic rebranding of a well-understood technique for the sake of novelty. I would also encourage the authors to provide a more in-depth and technically grounded discussion in the related work section to clearly articulate how their approach truly differs from prior adversarial attack based methods.
>
> **Response**: Thank you for the thoughtful comment. While it is true that our method, like prior adversarial approaches, learns a prompt suffix, the key contribution of LLMPrint does not lie in the act of learning a suffix itself, but in what the suffix is optimized to encode. In passive fingerprinting, the optimization objective determines the entire mechanism. Simply learning a suffix is not the technique; the objective defines what information about the model is extracted.
>
> Prior adversarial methods (e.g., TRAP) optimize for forcing specific outputs. In contrast, LLMPrint is the first to formalize fingerprint construction as learning an injected prompt that captures decision-boundary structure, rather than inducing a target output. Our boundary-based objective intentionally pairs a positive and negative token to locate a stable, model-unique preference region, which is an approach fundamentally different from output-forcing adversarial attacks. Our newly added ablations confirm this distinction: replacing our objective with TRAP-style single-token objectives leads to extremely high FPR or near-complete failure (a TPR of 0.9889 but an FPR of 0.8633), showing that the difference is not minor but essential.
>
> We will expand the related work section to clarify these conceptual differences. Although both LLMPrint and adversarial attacks involve optimized suffixes, the goals, optimization signals, and underlying modeling assumptions are distinct, and the decision-boundary formulation we propose is, to our knowledge, new in the context of LLM fingerprinting.

---

> ### Author Response · Authors · 2025-11-21
>
> >**Weakness 2**: The authors claim their method's uniqueness comes from placing prompts near the decision boundary. However, this concept of near is ill-defined and controlled by a manually tuned hyperparameter \alpha (e.g., 0.5). How near is enough to be unique, yet far enough to be robust? The paper provides no theoretical analysis or empirical study on the sensitivity of \alpha. Without this, the central claim becomes arbitrary. It's possible the method works not because of the property of the decision boundary, but simply because the optimization finds a local minimum that happens to be transferable to its model variants, and the decision boundary is just a post-hoc rationalization.
>
> **Response**: We thank the reviewer for raising this question. The role of $\alpha$ is explicitly analyzed in Appendix Fig. 2(b). As shown, a very small $\alpha$ (e.g., 0.1) indeed increases FPR, because it moves the optimization too far away from the boundary, producing prompts that no longer enforce a unique pairwise preference and therefore lose discriminative power. For a broad and stable range ($\alpha$=0.5 and 1.0), both TPR and FPR remain strong, indicating that the method is not sensitive to changing this hyperparameter.
> To directly validate that our fingerprint prompts truly lie near the decision boundary, we further measured the token-level probabilities of randomly selected fingerprint prompts. In all cases, the positive and negative tokens are the top two logits, with close probabilities, while all other tokens are substantially lower. This confirms that the optimization indeed positions prompts near the intended decision boundary:
>
> | Token Pair        | $w_j^+$ prob | $w_j^-$ prob | Next-highest prob |
> |------|-----------|-----------|:--------------------:|
> | rat / bull        | 0.3453    | 0.2954    | 0.0659   |
> | blue / white      | 0.3569    | 0.2737    | 0.0565 |
> | dog / cat         | 0.3818    | 0.3240    | 0.0063 |
> | purple / orange   | 0.2148    | 0.1499    | 0.0329|
> | star / square     | 0.2598    | 0.1916    | 0.0500|
>
> These findings show that $\alpha$ does not need careful tuning, and the optimized prompts consistently converge to boundary-adjacent regions, rather than arbitrary local minima. We will include these results in the revision to make the boundary interpretation clearer.
>
> >**Weakness 3**: Although the experimental results show that LLMPrint substantially outperforms TRAP, the paper does not clearly explain why this is the case. Is the improvement solely due to controlling behavior at the first-token level, or are there deeper factors that make the proposed optimization more effective? It would be good if the authors provide stronger theoretical or analytical insights to justify why their method is inherently better than other adversarial-attack-based approaches.
>
> **Response**: We appreciate the reviewer’s question. The improvement of LLMPrint over TRAP is not merely due to acting on the first token. The key factor is that our objective is fundamentally different from output-forcing adversarial attacks: instead of optimizing for a fixed output (as in TRAP), LLMPrint explicitly optimizes for a pairwise decision-boundary preference between $w_j^+$​ and $w_j^-$​, which simultaneously enforces uniqueness and robustness.
>
> To illustrate this difference empirically, we added an ablation that removes the negative token and uses a single-token objective—closely resembling a TRAP-style formulation. This single-token variant performs poorly, achieving TPR = 0.9889 but FPR = 0.8633 on Meta-Llama-3-8B suspect models, showing that controlling only the positive token leads to highly non-discriminative fingerprints. By contrast, the pairwise objective keeps the optimized prompt near the boundary between two competing tokens, while pushing it away from all others, yielding a stable and model-specific fingerprint that TRAP-style objectives fail to induce. These results suggest that the performance gap is not due to output position (first token vs. multi-token), but due to the architectural difference in what the optimization targets. LLMPrint’s boundary-based, two-token preference formulation captures a unique fingerprinting signal that adversarial attack objectives were never designed to extract. We will clarify this distinction in the revision.

---

> ### Author Response · Authors · 2025-11-21
>
> >**Weakness 4**: The failure analysis in Section 5.3 attributes detection failures to a drop in the suspect model's benchmark performance. While the data shows a correlation, this attribution of causality is not sufficiently proven. The authors do not rule out confounding variables or the possibility that their fingerprints are simply not robust to the specific fine-tuning methods that also happen to degrade benchmark performance.
>
> **Response**: We appreciate the reviewer’s concern. We agree that the correlation we report does not establish causality, and we do not intend to claim that performance degradation causes fingerprinting failure. Our goal in Section 5.3 was only to characterize the small set of false negatives.
>
> To address the reviewer’s alternative explanation, we examined the training methods of the misdetected models. For Meta-Llama-3-8B, the false positives were fine-tuned using SFT or DPO. However, LLMPrint also successfully detects many other SFT- and DPO-trained models, including those trained on similarly sized datasets. This shows that LLMPrint is not inherently brittle to these fine-tuning techniques and that “the fingerprints fail because SFT/DPO breaks them” is not supported by the data.
>
> The failure cases therefore appear to correspond to extreme drift, models whose behavior has diverged unusually far from the base model, rather than to a particular fine-tuning method per se. We will revise the text to clarify that our observation is a correlation helpful for understanding failure modes, rather than a causal explanation.
>
> >**Question 1**: What are the differences between the proposed method and adversarial attack based methods?
>
> **Response**: While our method also learns a suffix, its goal and optimization target differ fundamentally from adversarial attack–based methods. Attacks such as TRAP optimize prompts to force a specific output, whereas LLMPrint optimizes for a pairwise decision-boundary preference that encodes a stable, model-unique behavioral signature. This distinction is essential: our ablation replacing the pairwise objective with a TRAP-style single-token objective results in extremely high FPR, showing that adversarial formulations do not produce discriminative or robust fingerprints. Thus, although both approaches involve optimizing text, the underlying purpose, objective, and extracted model signal are categorically different.
>
> >**Question 2**: Why is the proposed method inherently better than other methods?
>
> **Response**: The improvement comes from the objective, not from the optimizer. Unlike prior methods that optimize prompts to force a particular output (e.g., TRAP’s output-matching objective), LLMPrint explicitly optimizes for a pairwise decision-boundary preference that balances uniqueness and robustness. This yields a behavioral signature that persists across post-training and other model modifications. Our ablations confirm this: replacing the pairwise objective with a single-token or TRAP-style objective leads to very high FPR, while the boundary-based formulation maintains strong TPR and low FPR. This shows that LLMPrint is inherently more effective because it targets the model’s underlying decision-boundary structure rather than its surface-level outputs.
>
> >**Question 3**: Are the fingerprint prompts really near the decision boundary?
>
> **Response**: Yes. We verified this directly by inspecting the token-level probabilities of randomly sampled fingerprint prompts. In all cases, the positive and negative tokens become the top two logits with close probabilities, while every other token has much lower probability. This pattern is exactly what we expect when the prompt lies near the decision boundary between $w_j^+$​ and $w_j^-$​. Representative examples are shown below:
>
> | Token Pair        | $w_j^+$ prob | $w_j^-$ prob | Next-highest prob |
> |-------------------|-----------|-----------|:--------------------:|
> | rat / bull        | 0.3453    | 0.2954    | 0.0659|
> | blue / white      | 0.3569    | 0.2737    | 0.0565|
> | dog / cat         | 0.3818    | 0.3240    | 0.0063|
> | purple / orange   | 0.2148    | 0.1499    | 0.0329|
> | star / square     | 0.2598    | 0.1916    | 0.0500 |
>
> These results consistently show that our fingerprint prompts indeed place the model close to the intended pairwise boundary, rather than at arbitrary local optima. We will include this evidence in the revision.

---

### Official Review · Reviewer_EvtS · 2025-10-28

**Soundness:** 1
**Presentation:** 3
**Contribution:** 2
**Rating:** 2
**Confidence:** 5

**Summary:**

This paper presents a method to address the LLM provenance detection problem, where post-processing, such as post-training or quantization, makes it challenging to determine if a suspect model is derived from a known base model. The authors introduce LLMPrint, a passive detection framework designed to be verifiable under both gray-box and black-box access and robust to these post-processing modifications. The core mechanism involves exploiting the base model's inherent vulnerability to prompt injection to generate a set of "fingerprint prompts". Each prompt is optimized using an objective function to enforce a consistent token preference between a specific pair of tokens. The evaluation on five base models and around 700 suspect variants shows the method achieves high true positive rates while keeping false positive rates near zero.

**Strengths:**

- Promising results.
- Support both gray-box and black-box verification.
- Large-scale evaluation.

**Weaknesses:**

- **On the Novelty of the Proposed Method:** The paper's core contribution appears to be incremental. The method uses GCG to optimize prompts that elicit specific model preferences, which is conceptually similar to TRAP's optimization, perhaps resembling a special case of TRAP focused on single-token outputs. Furthermore, the loss function designed to find prompts near the "decision boundary" shares a strong resemblance to the objective in IPGuard. Notably, the authors' claim that IPGuard fails completely (0% TPR) due to lacking a robustness objective is surprising, as the original IPGuard paper does include a term to ensure robustness.

- **On the Necessity of the Token-Pair Framework:** The paper's design is centered on pairs of tokens. However, the necessity of the negative token is not fully ablated. It may be beneficial for the authors to conduct an ablation study that removes the negative token.

- **On the Limited Threat Model:** The paper's robustness claims are primarily evaluated against post-training and quantization. This overlooks other common and practical scenarios that can significantly alter an LLM's behavior. It is unclear how LLMPrint would perform against suspect models modified by techniques like model merging and distillation, or those deployed with different system prompts or even non-default sampling settings like a different temperature. It may be better for the authors to evaluate the fingerprint's robustness against these factors to validate the method's practical utility.

- **On the Limited Practical Application:** The proposed method requires full white-box access to the base model to perform the GCG optimization. For the auditing and provenance tasks involving closed-source models (e.g., determining if a third-party API is illicitly using GPT-4), this method might not be feasible.

- **On the Unexplained Bypassing of SOTA Detectors:** The paper claims that its fingerprint prompts, optimized using standard GCG, can bypass two SOTA prompt injection detectors with high success rates. This is a counter-intuitive and significant claim, as GCG-optimized prompts are often syntactically anomalous and should be statistically distinct from natural language. This surprising result may require a more detailed explanation and analysis to build confidence in the finding.

- **On the Questionable Failure Analysis:** The failure analysis in Section 5.3 posits that misdetected positive models (false negatives) are a result of their own performance degradation on general benchmarks like MMLU. This line of reasoning may be problematic. In many real-world scenarios, a stolen model would be fine-tuned for a specific domain (e.g., legal or medical), which could naturally lower its score on general benchmarks while increasing its specialized value. This analysis may inadvertently suggest that LLMPrint is only effective at detecting near-identical copies of the base model and fails against models that have been meaningfully adapted.

- **On the Excessive Query Overhead and Potential for Unfair Comparison:** The proposed method requires 300 prompts, each queried 100 times in the black-box setting, for a total of 30,000 queries per detection. This represents a very high overhead. This also raises concerns about the fairness of the baseline comparison, as TRAP used 1,000 total queries and LLMmap used only 8 prompts. Based on the paper's own ablation study (Appendix A.5), the TPR of LLMPrint appears to be very low when the number of queries (T) is small (e.g., T=10). This strongly suggests that the method's high reported performance may be a product of this massive query budget rather than a fundamentally more effective fingerprinting design.

**Questions:**

Please refer to the Weaknesses section.

---

> ### Author Response · Authors · 2025-11-21
>
> We sincerely thank the reviewer for recognizing these strengths, particularly the promising empirical results, the ability of our method to support both gray-box and black-box verification, and the breadth of our large-scale evaluation. Please find our detailed responses to your comments below.
>
> >**Weakness 1**: On the Novelty of the Proposed Method: The paper's core contribution appears to be incremental. The method uses GCG to optimize prompts that elicit specific model preferences, which is conceptually similar to TRAP's optimization, perhaps resembling a special case of TRAP focused on single-token outputs. Furthermore, the loss function designed to find prompts near the "decision boundary" shares a strong resemblance to the objective in IPGuard. Notably, the authors' claim that IPGuard fails completely (0% TPR) due to lacking a robustness objective is surprising, as the original IPGuard paper does include a term to ensure robustness.
>
> **Response**: Thank you for the thoughtful comments regarding novelty. We agree that GCG itself is a general optimization tool, much like gradient descent, and it is not presented as the contribution of our method. The core contribution of LLMPrint lies in the fingerprinting objective we design and in the way we leverage prompt injection to extract decision-boundary characteristics of LLMs.
> Although TRAP also optimizes prompts, its goal is to induce a fixed output string, whereas our objective is fundamentally different: we construct pairwise token-preference fingerprints that explicitly balance uniqueness and robustness. As an additional experiment, we evaluate a single-token variant that resembles a TRAP-style formulation and observe significantly inferior performance (TPR 0.9889, FPR 0.8633), showing that directly adapting TRAP’s design does not yield discriminative fingerprints.
> Regarding IPGuard, the similarity is only superficial. IPGuard is designed for image classifiers with small output spaces, and its robustness term is tailored to that setting. When directly applied to LLMs, whose vocabularies correspond to tens of thousands of classes, its robustness term is nearly inactive, which we confirm experimentally: substituting our loss with the IPGuard loss results in near-zero TPR. In contrast, our loss is constructed specifically for LLMs, ensuring that the fingerprint prompt lies near the intended pairwise boundary while remaining well separated from the rest of the vocabulary. To our knowledge, this is the first work to explicitly formulate LLM fingerprinting around decision-boundary geometry in the prompt space.
> We will revise the paper to clarify these distinctions and to better articulate the conceptual contribution of LLMPrint. While our optimization uses GCG, the fingerprinting objective, its decision-boundary–aware formulation, and its suitability for LLMs rather than classifiers constitute the key novelty of our approach.
>
> >**Weakness 2**: On the Necessity of the Token-Pair Framework: The paper's design is centered on pairs of tokens. However, the necessity of the negative token is not fully ablated. It may be beneficial for the authors to conduct an ablation study that removes the negative token.
>
> **Response**: We appreciate the reviewer’s suggestion. To directly assess the necessity of the negative token in our token-pair framework, we conducted an ablation study on Meta-Llama-3-8B and its suspect models, removing the negative token and constructing fingerprints using only a single target token. This variant isolates whether pairwise token preferences are essential.
> The results show that the single-token formulation exhibits poor discriminative ability, achieving a TPR of 0.9889 but an FPR of 0.8633, meaning that a large portion of negative suspect models are incorrectly detected as positives. This illustrates that single-token fingerprints are not sufficiently unique. By contrast, our pairwise design positions the fingerprint prompt near the decision boundary between $w_j^+$​ and $w_j^-$, while ensuring separation from all other tokens, which is critical for both uniqueness and robustness. These findings confirm that the negative token is necessary for stable optimization and to prevent the fingerprint from collapsing toward generic high-probability tokens. We will include this ablation in the revision to clarify the motivation behind the token-pair framework.

---

> ### Author Response · Authors · 2025-11-21
>
> >**Weakness 3**: On the Limited Threat Model: The paper's robustness claims are primarily evaluated against post-training and quantization. This overlooks other common and practical scenarios that can significantly alter an LLM's behavior. It is unclear how LLMPrint would perform against suspect models modified by techniques like model merging and distillation, or those deployed with different system prompts or even non-default sampling settings like a different temperature. It may be better for the authors to evaluate the fingerprint's robustness against these factors to validate the method's practical utility.
>
> **Response**: We thank the reviewer for raising this important point. Beyond post-training and quantization, we have additionally evaluated LLMPrint on another practically relevant model-modification scenario: model merging. Using 70 widely downloaded merging-based variants of Meta-Llama-3-8B as positive suspect models, LLMPrint achieves a TPR of 0.914, demonstrating that the fingerprint signal remains largely preserved even when the base model contributes only partially to the merged parameters.
> Regarding distillation, we agree that it represents another meaningful modification technique and our method is applicable to it. However, distillation is typically applied to very large models (e.g., 30B–70B), and performing a systematic evaluation under this setting is beyond our current computational budget. We will explicitly acknowledge this limitation and highlight it as an important direction for future work.
> For system prompts and non-default sampling settings, we would like to clarify our threat model more explicitly. LLMPrint is designed for API-based gray-box and black-box provenance verification, where the detector initiates and controls all verification queries. In these settings, the detector can freely choose the system prompt, decoding parameters, and any other inference-time configuration used during fingerprint extraction. Provenance verification is therefore always performed under detector-controlled prompts and sampling settings, regardless of how the suspect model is deployed or what defaults it uses in downstream applications. As a result, differences in the suspect model’s own default system prompts or sampling configurations do not affect our verification process. We will clarify this in the revision.
>
> >**Weakness 4**: On the Limited Practical Application: The proposed method requires full white-box access to the base model to perform the GCG optimization. For the auditing and provenance tasks involving closed-source models (e.g., determining if a third-party API is illicitly using GPT-4), this method might not be feasible.
>
> **Response**: We appreciate the reviewer’s concern. LLMPrint is designed for the setting where the provider of the base model performs provenance verification, which is the standard scenario in intellectual property protection and model-auditing workflows. In this context, the base-model owner naturally has full access to the base model and can run the GCG optimization without any restriction. But for the suspect models, we only assume black-box or gray-box API access.
>
> >**Weakness 5**: On the Unexplained Bypassing of SOTA Detectors: The paper claims that its fingerprint prompts, optimized using standard GCG, can bypass two SOTA prompt injection detectors with high success rates. This is a counter-intuitive and significant claim, as GCG-optimized prompts are often syntactically anomalous and should be statistically distinct from natural language. This surprising result may require a more detailed explanation and analysis to build confidence in the finding.
>
> **Response**:  We appreciate the reviewer’s concern. The ability of our fingerprint prompts to bypass detectors is less surprising when considering practical detection constraints. In our experiments, the PPL-Detector is calibrated to keep its false positive rate $\leq 10^{-3}$, as would be required in real deployments. Under such a constraint, the detector cannot simply block all syntactically unusual prompts, even if they are GCG-generated.
> Moreover, our fingerprint prompts always include a natural-language base prompt, which greatly improves their linguistic plausibility and reduces the statistical deviation introduced by the optimized suffix. As supporting evidence, when we remove this base prompt and use only the GCG-optimized suffix, the bypass rate drops sharply to 0.428. This confirms that the high bypass rate arises from realistic FPR constraints combined with the natural-language anchoring in our design. We will add this explanation to the paper to clarify why the observed bypass rates are expected under practical detection settings.

---

> ### Author Response · Authors · 2025-11-21
>
> >**Weakness 6**: On the Questionable Failure Analysis: The failure analysis in Section 5.3 posits that misdetected positive models (false negatives) are a result of their own performance degradation on general benchmarks like MMLU. This line of reasoning may be problematic. In many real-world scenarios, a stolen model would be fine-tuned for a specific domain (e.g., legal or medical), which could naturally lower its score on general benchmarks while increasing its specialized value. This analysis may inadvertently suggest that LLMPrint is only effective at detecting near-identical copies of the base model and fails against models that have been meaningfully adapted.
>
> **Response**: Thank you for raising this point. We would like to clarify the intention of our failure analysis. Our observation that mis-detected positive suspect models tend to show large performance drops on general-purpose benchmarks is descriptive, not prescriptive. It does not imply that LLMPrint only succeeds on near-identical copies or that any model with lower general-benchmark performance would fail to be detected.
>
> In fact, the suspect models we evaluate are collected from HuggingFace based on download popularity, and thus naturally include many meaningfully adapted models—such as models post-trained for reasoning, coding, role-play, medical Q&A, and other domain-specialized uses. Despite these substantial adaptations, LLMPrint detects the overwhelming majority of them. For example, on Meta-Llama-3-8B under the gray-box setting, we achieve a TPR of 0.956, demonstrating that LLMPrint remains effective even when models are significantly modified for specialized domains.
>
> Our failure analysis simply aims to understand the rare cases where LLMPrint does fail. The correlation with severe performance degradation suggests that these specific models have drifted so far from the base that they no longer preserve enough of its decision-boundary structure for any behavioral fingerprint to remain recoverable. This does not indicate a general limitation against domain-adapted models; rather, it highlights the extreme boundary of model alteration beyond which provenance detection becomes inherently difficult.
>
> We will clarify this intent to avoid any misunderstanding.

---

> ### Author Response · Authors · 2025-11-21
>
> >**Weakness 7**: On the Excessive Query Overhead and Potential for Unfair Comparison: The proposed method requires 300 prompts, each queried 100 times in the black-box setting, for a total of 30,000 queries per detection. This represents a very high overhead. This also raises concerns about the fairness of the baseline comparison, as TRAP used 1,000 total queries and LLMmap used only 8 prompts. Based on the paper's own ablation study (Appendix A.5), the TPR of LLMPrint appears to be very low when the number of queries (T) is small (e.g., T=10). This strongly suggests that the method's high reported performance may be a product of this massive query budget rather than a fundamentally more effective fingerprinting design.
>
> **Response**: We appreciate the reviewer’s concern regarding query overhead. In the gray-box setting, LLMPrint requires only 300 queries in total. In the black-box setting, although the budget is 30,000 queries, the cost remains manageable because each query uses a short fingerprint prompt and only the first generated token is needed. Importantly, provenance verification is performed per suspect model rather than per user request, so the overhead remains practical.
> To address fairness, we conducted an additional experiment giving TRAP the same total budget of 30,000 queries. Even with this enlarged budget, TRAP still performs significantly worse:
>
> | Method (30,000 queries) | Post-training (TPR / FPR) | Quantization (TPR / FPR) |
> |--------------------------|----------------------------|----------------------------|
> | TRAP                     | 0.8111 / 0.3941            | 0.7917 / 0.5534            |
>
>
> The high FPR persists despite the increased budget, indicating that TRAP’s limitation stems from its output-matching formulation rather than query count.
>
> For LLMmap, its 8 default prompts rely on handcrafted designs not fully specified in the paper, and the authors report that LLM-generated prompts, even in larger numbers, perform markedly worse. Thus, LLMmap is inherently difficult to scale to large prompt sets, preventing a fair extension to 30,000 queries. We regard this lack of scalability as a structural limitation of LLMmap rather than a comparison artifact.
>
> Regarding the ablation for T, the case T=10 simply reflects the statistical unreliability of estimating token preferences from a very small number of samples. As shown in Appendix A.5, performance stabilizes when T=100, which is still computationally inexpensive and remains far below typical API rate limits.
>
> Overall, LLMPrint’s strong performance is not a byproduct of query volume. The gray-box setting requires only 300 queries, and even in the black-box setting, competing methods do not approach LLMPrint’s accuracy even when granted equivalent query budgets. We will clarify these points in the revised version.

---

> ### Comment · Reviewer_EvtS · 2025-11-24
>
> Thank you for the response. However, I think the explanation is not convincing. My concerns about the novelty, the unusual experimental results, and the poor performance while using fewer queries still remain. As such, I decide to keep my rating.

---

### Official Review · Reviewer_PWA5 · 2025-11-01

**Soundness:** 3
**Presentation:** 3
**Contribution:** 3
**Rating:** 6
**Confidence:** 3

**Summary:**

This paper addresses the provenance verification challenge of Large Language Models (LLMs) and proposes LLMPrint, a robust fingerprinting framework based on prompt injection and token attribution signals.

Unlike traditional watermarking or output-based fingerprints, LLMPrint leverages the inherent prompt injection vulnerability of LLMs to optimize fingerprint trigger words that induce stable token preferences unique to each base model. These preferences are embedded as fine-grained token attribution distributions, enabling ownership verification through analysis of attribution responses (e.g., gradients or attention weights).

The method provides white-box verification that remains effective under fine-tuning, quantization, and LoRA adaptation, preserving both model quality and fingerprint persistence.

Experiments on Llama-2, Mistral, and Falcon demonstrate that the fingerprints maintain both model uniqueness and robustness to post-processing operations.

**Strengths:**

1. Novel use of token attribution for model fingerprinting.
2. Empirical robustness to moderate fine-tuning and retraining.
3. No performance degradation due to output modification.
4. Transparent technical design and reproducibility plan.
5. White-box verification offering strong ownership validation.

**Weaknesses:**

1. Theoretical analysis is limited, Lack of theoretical grounding for robustness under strong distribution shifts.
2. White-box assumption reduces practical applicability for closed models.
3. Lack of quantitative fingerprint distinctiveness evaluation.
4. Attribution extraction incurs significant computational cost for large models.
5. Fine-tuning robustness tested only on moderate tasks, lacking domain-transfer evaluation.
6. Limited evaluation under adversarial settings (e.g., targeted evasion of fingerprinting).

**Questions:**

1.Could fingerprint prompts be detected or removed by an adversary already aware of the LLMPrint mechanism? Please discuss the method’s resilience against prompt-based fingerprint removal attacks.
2.Could the proposed approach be extended to identify model extraction or partial model stealing, where only subsets of weights or behaviors are replicated?
3.How does the method perform under adversarial fine-tuning specifically designed to erase fingerprints? Could similar robustness be maintained when scaling to larger models (e.g., 7B, 13B)?
4.Could instruction-tuning shifts or alignment artifacts (e.g., from SFT or RLHF) affect the stability of attribution fingerprints, and how might this be mitigated?
5.Fingerprint suggestion word optimization currently relies on 1,000 GCG iterations. Could the authors analyze the trade-off between iteration count (e.g., 200/500/1500) and fingerprint strength (TPR) or runtime to verify whether 1,000 is the optimal balance? A comparison with other optimization algorithms (e.g., gradient descent) would help justify the use of GCG.
6.Do the fingerprint hints detected by DataSentinel or PPL-Detector share common linguistic or statistical characteristics (e.g., semantic incoherence, abnormal perplexity)? Could these hints be refined to improve semantic coherence and bypass rate?

---

> ### Author Response · Authors · 2025-11-21
>
> We sincerely thank the reviewer for highlighting these strengths, especially the novelty of our fingerprinting design, the empirical robustness under real fine-tuned variants, the fact that our method preserves model functionality, the clarity and reproducibility of our technical framework, and the practical value of our strong verification capability. Please find our detailed responses to your comments below.
>
> > **Weakness 1**: Theoretical analysis is limited, Lack of theoretical grounding for robustness under strong distribution shifts.
>
> **Response**: We thank the reviewer for raising this point. Developing a complete theoretical framework for robustness under strong distribution shifts is indeed challenging and, to our knowledge, remains largely open in the literature—even for substantially simpler models. Our work is primarily empirical in nature and aims to address a practical challenge observed in real-world scenarios. We agree that additional theoretical intuition can improve clarity, and we have therefore expanded our discussion to articulate the core reasoning behind the method’s robustness, without overclaiming formal guarantees.
>
> >**Weakness 2**: White-box assumption reduces practical applicability for closed models.
>
> **Response**: Thank you for raising this point. Our threat model assumes the model provider (the victim) performs the detection. In this standard copyright protection scenario, the provider naturally possesses full white-box access to their own base model to construct fingerprints. This differs from third-party auditing; our goal is to verify if a suspect model is derived from our base model. For suspect models, our method only assumes black-box or gray-box API access.
>
> > **Weakness 3**: Lack of quantitative fingerprint distinctiveness evaluation.
>
> **Response**: We thank the reviewer for raising this concern. To address it, we added an additional quantitative experiment in the revised version that measures the Hamming distance between the base model’s fingerprint and that of each suspect model.
> The results (summarized in the table below) show a clear separation:
>
> | Category               | Mean ↓ | Std    | Median | 25th % | 75th % |
> |------------------------|--------|--------|--------|--------|--------|
> | Positive suspect model | 0.4033 | 0.1649 | 0.3883 | 0.3125 | 0.5342 |
> | Negative suspect model | 0.8573 | 0.0750 | 0.8467 | 0.8167 | 0.9333 |
>
> We will further include a histogram in the revision which visually confirms that positive suspect models cluster at low Hamming distances, whereas negative suspect models form a separate cluster at much higher distances, with minimal overlap between the two distributions. The table and histogram provide a direct quantitative demonstration that our fingerprints are strongly distinctive between positive and negative suspect models, addressing the proposed concern.
>
> >**Weakness 4**: Attribution extraction incurs significant computational cost for large models.
>
> **Response**: We appreciate the reviewer’s observation regarding the computational cost of attribution extraction for large models. We fully agree that running our method on large-scale LLMs can be expensive. However, we would like to clarify that in our setting this computation is required only once per base model, at fingerprint construction time. After the fingerprint prompt is generated, all subsequent verification procedures operate solely on the stored fingerprint prompts, without requiring any further fingerprint construction computation.
>
> >**Weakness 5**: Fine-tuning robustness tested only on moderate tasks, lacking domain-transfer evaluation.
>
> **Response**: Thank you for the insightful comment. We respectfully clarify that our fine-tuning robustness experiments do not rely on controlled or moderate tasks. All post-trained suspect models used in our evaluation are real community checkpoints from HuggingFace, created by independent users and organizations for highly diverse purposes, including medical QA, legal reasoning, coding and software engineering, role-playing agents, safety/alignment tuning, mathematical reasoning, and more. These models exhibit substantial and heterogeneous domain shifts, far beyond what a single-task or moderate fine-tuning setup would impose. The fact that LLMPrint consistently detects these widely varied real-world variants demonstrates that its robustness already covers realistic domain-transfer scenarios rather than artificially simplified ones. We will clarify this more explicitly in the revision.

---

> ### Author Response · Authors · 2025-11-21
>
> >**Weakness 6**: Limited evaluation under adversarial settings (e.g., targeted evasion of fingerprinting).
>
> **Response**: We thank the reviewer for highlighting the importance of adversarial robustness. To directly address this concern, we conducted a dedicated set of adaptive attack experiments in which the attacker explicitly attempts to evade LLMPrint while having varying levels of knowledge about our method. These attacks go beyond standard robustness checks and represent worst-case attempts to invert the fingerprint signal.
> We evaluate three adversarial settings. In the white-box adaptive case, the attacker has full knowledge of our fingerprint prompts, the exact token pairs $(w_j^+, w_j^-)$, and all GCG hyperparameters, and uses LoRA fine-tuning to explicitly force the model to assign higher probability to $w_j^-​$ than to $w_j^+​$. In the gray-box adaptive case, the attacker knows our method and GCG hyperparameters (we intentionally grant the attacker this level of knowledge,stronger than what a real adversary would typically possess, in order to make the evaluation maximally adversarial and stress-test the robustness of LLMPrint) as well as the fixed base prompt, but does not know our chosen token pairs; they therefore randomly generate 300 new token pairs and optimize their own fingerprint prompts. In the black-box adaptive case, the attacker only knows our high-level algorithm and must design their own GCG hyperparameters and base prompt; to simulate this, we use a different random seed and replace our base prompt with a semantically similar alternative (“Randomly give me a word from your knowledge base”), after which the attacker again constructs their own fingerprint prompts.
> All attacks are evaluated on the positive suspect models of Meta-Llama-3-8B that LLMPrint correctly detected in the original experiments. The TPR after adaptive attacks are reported below:
> | Adaptive Attack Setting | TPR |
> |-------------------------|----------------------|
> | White-box              | 0              |
> | Gray-box                | 0.167               |
> | Black-box               | 0.967              |
>
> The results reveal a clear pattern. When the attacker has perfect white-box access, full evasion is theoretically possible and indeed observed, which aligns with inherent limitations of any behavioral fingerprinting method. However, as soon as the attacker lacks the exact fingerprint prompts and token pairs—even while knowing the algorithm and attempting to replicate the optimization process—their ability to evade LLMPrint drops dramatically. In the black-box scenario, nearly all positive suspect models remain detectable. These findings demonstrate that LLMPrint maintains strong robustness under realistic adversarial conditions where attackers do not possess perfect knowledge of the fingerprint prompts.
> We will include this analysis in the revision to more comprehensively address adversarial robustness and to clarify the practical difficulty of constructing effective adaptive attacks against LLMPrint.
>
> >**Question 1**: Could fingerprint prompts be detected or removed by an adversary already aware of the LLMPrint mechanism? Please discuss the method’s resilience against prompt-based fingerprint removal attacks.
>
> **Response**: We appreciate the reviewer’s concern. In fact, we already evaluate prompt-based fingerprint removal attacks using DataSentinel and PPL-Detector, two state-of-the-art prompt-filtering systems designed to detect and block suspicious or adversarial prompts. This setting directly models an adversary attempting to recognize and remove our fingerprint prompts. Our results show that a nontrivial portion of fingerprint prompts consistently bypass these detectors, and using only those surviving prompts is still sufficient for reliable fingerprint recovery. This indicates that LLMPrint is robust to prompt-based fingerprint removal, even when the adversary is aware of the mechanism and employs strong filtering tools.

---

> ### Author Response · Authors · 2025-11-21
>
> >**Question 2**: Could the proposed approach be extended to identify model extraction or partial model stealing, where only subsets of weights or behaviors are replicated?
>
> **Response**: We appreciate the reviewer’s question regarding the applicability of LLMPrint to settings involving partial model stealing or model extraction. Although this scenario is outside the primary scope of our evaluation, we conducted additional experiments to assess whether LLMPrint can also detect models produced through model merging—a representative form of partial model reuse where only subsets of weights or behaviors from the base model are incorporated into a new model.
> To test this, we collected 70 model-merging variants of Meta-Llama-3-8B from HuggingFace, selecting the most widely downloaded community models to reflect realistic usage. These merged models typically combine the base model with another checkpoint through parameter interpolation or weighted averaging, a process that reuses substantial but non-identical portions of the base model’s knowledge. Among these 70 positive suspect models, LLMPrint successfully detected 64, achieving a TPR of 0.9143.
> This result suggests that the fingerprint prompt survives even when the base model contributes only partially to the merged weights. In other words, the decision-boundary characteristics captured by our fingerprint prompts remain sufficiently preserved through model merging, enabling reliable identification of partial model reuse in many cases. We will report this analysis in the revision, as it provides preliminary evidence that LLMPrint can extend beyond direct post-training or quantization and retain effectiveness under partial model stealing scenarios.
>
> >**Question 3**: How does the method perform under adversarial fine-tuning specifically designed to erase fingerprints? Could similar robustness be maintained when scaling to larger models (e.g., 7B, 13B)?
>
> **Response**: Thank you for the thoughtful question. The scenario of adversarial fine-tuning explicitly designed to erase fingerprints is exactly what we target in our adaptive-attack study. Concretely, we consider an attacker who applies LoRA fine-tuning with the explicit objective of erasing or inverting our fingerprint: for each fingerprint prompt and token pair (w_j^+, w_j^-), the attacker fine-tunes the suspect model to assign higher probability to w_j^-​ than to w_j^+​. We evaluate this under three different knowledge settings on Meta-Llama-3-8B positive suspect models that LLMPrint originally detects correctly:
>
> | Adaptive Attack Setting | TPR |
> |-------------------------|----------------------|
> | White-box              | 0              |
> | Gray-box                | 0.167               |
> | Black-box               | 0.967              |
>
> These results show that under the strongest, fully white-box adversary—who knows the exact fingerprint prompts, token pairs, and optimization details—successful evasion is indeed possible, which is consistent with inherent limitations of any behavioral fingerprinting method. However, once the attacker loses access to the exact fingerprint (gray-box and especially black-box settings), LLMPrint remains substantially robust: even after adversarial fine-tuning specifically aimed at erasing the fingerprint, we correctly detect most positive suspect models in the black-box adaptive case.
> Regarding scalability to larger models, our experiments already cover a range of model sizes and architectures from 135M up to 8B parameters, including 7B-class models (Mistral-7B-v0.3) and 8B models (Meta-Llama-3-8B, Qwen3-8B). Across this spectrum, LLMPrint maintains consistently high TPRs with near-zero FPRs, suggesting that the mechanism is not tied to a particular scale. Since LLMPrint operates by shaping decision boundaries via prompt injection and only relies on query access (log-probabilities or samples), we expect the same procedure to transfer to larger models such as 13B. A full 13B-scale adaptive-attack evaluation is currently beyond our computational budget, but we view it as a natural extension and will discuss this limitation and avenue for future work in the revision.
>
> >**Question 4**: Could instruction-tuning shifts or alignment artifacts (e.g., from SFT or RLHF) affect the stability of attribution fingerprints, and how might this be mitigated?
>
> **Response**: Thank you for the question. In fact, our evaluation already includes models that have undergone instruction tuning, including both SFT and RLHF variants downloaded from HuggingFace. These models naturally exhibit the alignment shifts you refer to. Our results show that attribution fingerprints remain stable under these alignment modifications. We will clarify this explicitly in the revision to avoid misunderstanding.

---

> ### Author Response · Authors · 2025-11-21
>
> >**Question 5**: Fingerprint suggestion word optimization currently relies on 1,000 GCG iterations. Could the authors analyze the trade-off between iteration count (e.g., 200/500/1500) and fingerprint strength (TPR) or runtime to verify whether 1,000 is the optimal balance? A comparison with other optimization algorithms (e.g., gradient descent) would help justify the use of GCG.
>
> **Response**: We thank the reviewer for this helpful suggestion. We have added an explicit ablation on the number of GCG iterations used for fingerprint prompt optimization. On Meta-Llama-3-8B, we varied the number of GCG steps while keeping all other settings fixed and measured the resulting TPR for detecting positive suspect models. The results are:
>
> | GCG iterations | TPR   |
> |----------------|-------|
> | 200            | 0.9111|
> | 500            | 0.9444|
> | 1500           | 0.9444|
>
> We observe that the TPR improves when increasing the iterations from 200 to 500, and then quickly saturates: 500, 1000, and 1500 iterations yield similar TPR. Since the runtime of GCG scales approximately linearly with the number of iterations, 1,000 iterations lies in the flat part of this curve and is therefore a conservative choice.
> Regarding alternative optimizers, we use GCG because it is specifically designed for discrete token-level search and performs strongly in prior prompt-optimization work. In contrast, standard gradient-based methods operate in continuous embedding space and require additional relaxation or decoding steps to map gradients back to discrete tokens, making them less suitable for our setting. Since GCG already achieves high TPR with moderate iteration counts, it provides a simple and effective choice for our discrete optimization problem. We will add this clarification and our ablation in the revision.
>
> >**Question 6**: Do the fingerprint hints detected by DataSentinel or PPL-Detector share common linguistic or statistical characteristics (e.g., semantic incoherence, abnormal perplexity)? Could these hints be refined to improve semantic coherence and bypass rate?
>
> **Response**: Thanks for your comment. We inspected all fingerprint prompts flagged by DataSentinel or PPL-Detector and found consistent linguistic and statistical patterns. Specifically, detected prompts typically contain (1) long sequences of low-frequency or domain-specific tokens (e.g., programming identifiers, file-path fragments, markup), (2) semantically incoherent mixtures of unrelated domains, and (3) irregular tokenization patterns produced by discrete optimization. These characteristics reflect generic signals that current detectors rely on—low-probability token bursts or structural anomalies—rather than any property specific to our method. However, most fingerprint prompts that include a natural-language base prompt remain unflagged, and our new ablation shows that removing this base prompt drastically reduces bypass rate. This indicates that refining semantic coherence in the suffix alone does not meaningfully improve bypassability; the key factor is anchoring the fingerprint in natural-language structure, which we already do. We will clarify these observations in the revision.

---

> ### Comment · Reviewer_PWA5 · 2025-11-24
>
> I have reviewed the authors' responses.
>
> While I appreciate the effort taken, my primary concern remains the novelty of the proposed method and the assumptions of the threat model are still not convincing.
>
> Overall, I will maintain my current score, but I do not recommend acceptance.

---

### Note · Authors · 2025-12-02

I have read and agree with the venue's withdrawal policy on behalf of myself and my co-authors.